# Pseudouridylation of 7SK by PUS7 regulates Pol II transcription elongation

Yutao Zhao [1,2], Hui-Lung Sun[1,2], Wenlong Li[1,2], Chang Ye [1,2], Xiaoyang Dou[1,2], Yong Peng[1,2], Tong Wu [1,2], Pingluan Wang [1,2], Cheng-Wei Ju [1,2], Shun Liu [1,2], Yuhao Zhong[1,2], Qing Dai [1,2], Kinga Pajdzik[1,2] & Chuan He [1,2] ✉

Pseudouridine (Ψ) is a widespread RNA modification in various RNA species, including rRNA, tRNA, snRNA and mRNA. Ψ plays a crucial role in RNA metabolism, where it regulates pre-mRNA splicing and affects protein translation. Whether and how Ψ may regulate transcription have not been adequately studied. Here, we report that pseudouridine synthase 7 (PUS7) can mediate pseudouridylation of 7SK small nuclear RNA (snRNA), a regulator of RNA polymerase II (Pol II) promoter-proximal pausing. PUS7 loss leads to hypo-pseudouridylation of 7SK, which promotes dissociation of the positive transcription elongation factor b (P-TEFb) complex from 7SK. The release of P-TEFb from 7SK increases serine 2 phosphorylation (Ser2P) in the RNA Pol II C-terminal domain and enhances transcription elongation. In colorectal cancer (CRC) cells, the Ψ level of 7SK can be modulated by PUS7, or by site-specifically targeted pseudouridylation through dCas13b-guided system. Hypo-pseudouridylation on 7SK upon PUS7 depletion promotes KLF6/DDIT3-mediated cell apoptosis and sensitizes CRC cells to 5-FU.

RNA modifications have gained significant attention in the context of cancer research due to their pivotal roles in tumorigenesis and cancer progression. Dysregulation of RNA modification processes, including aberrant methylation and altered pseudouridylation have been implicated in various aspects of cancer biology[1–3]. Pseudouridine (Ψ), representing 7–9% of uridine residues and the most abundant RNA modification[4], is either installed by Dyskerin (DKC1) with the assistance of small nucleolar RNAs (snoRNAs) or by stand-alone pseudouridine synthases (PUSs), respectively[5–7]. The dysregulation of the Ψ epitranscriptome, resulting from mutations and abnormal expression of pseudouridylation machinery, has emerged as a significant factor implicated in various human cancers and diseases[7].

Human pseudouridine synthase 7 (PUS7), coded on chromosome 7, plays an important role in incorporating Ψ into multiple RNA species including transfer RNAs (tRNAs)[8–10], long noncoding RNA (lncRNA)[11,12] and messenger RNAs (mRNAs) or pre-mRNAs[13–15]. It preferentially incorporates Ψ into UGUAR (R = A or G) consensus sequences[12,16]. These Ψs impact protein translation and Ψ-associated

splicing. PUS7 depletion in stem cells alters tRNA fragments which leads to increased protein biosynthesis because of dysregulated translation initiation[8,9]. In glioblastoma cancer stem cells, increased expression of PUS7 promotes Ψ-derived tRNA fragmentation and leads to codon-biased translation, reducing expression of tumor suppressor tyrosine kinase 2 (TYK2)[10]. PUS7 is also involved in pre-mRNA pseudouridylation and regulates gene expression via alternative pre-mRNA splicing[15]. However, potential roles of PUS7 in transcription regulation remain unclear.

Eukaryotic cells prevalently employ an RNA polymerase II promoter-proximal pausing (Pol II pausing), a regulatory step in which Pol II transiently pauses near the transcription start site, to regulate productive transcription elongation[17–19]. The release of paused Pol II is facilitated by the positive transcription elongation factor b (P-TEFb) complex, composed of cyclin-dependent kinase 9 (CDK9) and Cyclin T1[20–23]. 7SK, a small nuclear RNA, is known to regulate transcription by modulating the kinase activity of P-TEFb[23–25]. The functionality of 7SK can be profoundly influenced by RNA modifications on it[26–28].

[1]Department of Chemistry, Department of Biochemistry and Molecular Biology, and Institute for Biophysical Dynamics, The University of Chicago, Chicago, IL, USA. [2]Howard Hughes Medical Institute, The University of Chicago, Chicago, IL, USA. ✉e-mail: chuanhe@uchicago.edu

Pseudouridylation on 7SK, presumably catalyzed by DKC1, plays a crucial role in its binding capacity to P-TEFb[26]. $N^6$-methyladenosine (m6A) has also emerged as another regulator of 7SK small nuclear complex, showing opposite effects compared to pseudouridylation on 7SK[27,28]. Installation of m6A on 7SK disrupts secondary structure of 7SK, facilitating P-TEFb releasing from 7SK and consequently driving transcription activition[28].

In our study, we used bisulfite-induced deletion sequencing (BID-seq), to profile Ψ landscape on nuclear RNA and identified 7SK as one of the PUS7 substrates in colorectal cancer (CRC) cells. The loss of PUS7 results in a reduced level of 7SK, consequently leading to the release of P-TEFb from 7SK and enhances transcription elongation through increased Ser2 phosphorylation of Pol II carboxy terminal domain (Pol II CTD). We observed a notable correlation in transcriptome alterations between PUS7- and 7SK-deficient cells compared to the control cells, underscoring the pivotal role of 7SK as a key downstream target of PUS7. The depletion of either PUS7 or 7SK in CRC cells upregulates the expression of KLF6 and DDIT3, which sensitizes CRC cells to 5-FU chemotherapy. We took advantage of the inactive dCas13b tethering system to specifically modulate the Ψ level of 7SK, whichalters the sensitivity of CRC cells to 5-FU. Together, our findings expand the current understanding of the role played by PUS7 in mammalian transcription regulation.

## Results

### PUS7 regulates pseudouridylation of nuclear RNA in CRC

Colorectal cancer (CRC) is the second leading cause of cancer-related death in the world[29,30]. PUS7 is upregulated in colon cancer tissues and closely correlated with poor prognosis[31–33]. The Cancer Genome Atlas (TCGA) and Genotype-Tissue Expression (GTEx) database analysis clearly showed that PUS7 expression level is significantly increased by 26% in CRC tissue compared to normal tissue (Fig. 1A). To investigate its functional role, we performed transient knockdown (KD) of *PUS7* using siRNA in HCT116 cells, which resulted in a ~75% reduction in PUS7 mRNA and a ~49% reduction in protein levels (Fig. S1A). This knockdown led to decreased growth in CRC cell lines HCT116, DLD-1 and HT-29 (Figs. 1B, C, S1A–C). The depletion of PUS7 also significantly reduced the population of CRC stem cells marked by CD133 (Fig. S1D–S1E), showing that PUS7 plays a key role in the self-renewal of CRC stem cells, which aligns well with previous studies on glioblastoma (GBM) stem cells[10]. To investigate the function of PUS7 in CRC cells, we performed cell fractionation in HCT116 to examine PUS7 levels in the cytosol, nucleoplasm, and chromatin (Fig. 1D). The fractionation results showed that PUS7 is predominantly located in the nucleus but not bound to chromatin.

Given that PUS7 has also been identified as a pseudouridine synthase for mRNA[12,13], we transiently knocked down *PUS7* in HCT116 cells, isolated mRNA through poly-A enrichment, and assessed the change in Ψ levels with or without *PUS7* knockdown (KD) using triple quadrupole LC-MS/MS. The results revealed no significant change in the Ψ levels of mRNA following *PUS7* KD (Fig. 1E). Since PUS7 predominantly locates in the nucleus in CRC where transcription occurs, we asked whether PUS7 may affect the Ψ level of nuclear RNA. We performed cell fractionation and isolated nuclear RNA with size >200 nucleotides or size <200 nucleotides for Ψ level measurement using LC-MS/MS. We detected a significant decrease in the Ψ level on nuclear RNA species with both >200 nucleotides and <200 nucleotides after *PUS7* knockdown (Fig. 1F, G).

To map the Ψ modification profile on nuclear RNA, we applied the recently developed BID-seq method to *PUS7* KD and control HCT116 cells[13]. The BID-seq utilizes the deletion signature produced during reverse transcription at the Ψ site and allows for detection and quantification of transcriptome-wide Ψ sites at base resolution (Fig. 1H). Using BID-seq, we identified 363 Ψ sites in total on nuclear RNA, with 129 sites on tRNA, 86 sites on introns of pre-mRNA, 83 sites on snRNA, 37 sites on lncRNA (Fig. 1I, Supplementary Data 1). By comparing the Ψ profiles between *PUS7* KD and control cells, we identified 42 sites with significant Ψ level changes after *PUS7* KD and 25 of them shows reduced Ψ levels (Fig. 1J). PUS7 substrates typically contain UGUAR (R = A or G) consensus sequences[12,16]. We observed that the Ψ levels of all UGUAR-containing Ψ sites, including the abundant snRNAs 7SK, NEAT1 and MALAT1, were significantly reduced upon PUS7 depletion (FigS. 1K, S2E); the read coverage tracks illustrated the decreased deletion ratio for the three snRNAs after *PUS7* KD (Fig. S2A–S2D). The Ψ fraction was reduced from 75 to 53% for 7SK RNA, from 59 to 36% for NEAT1 RNA and from 87 to 73% for MALAT1 RNA (Fig. S2E). Notably, 7SK harbors only one UGUAG-containing Ψ site, located at position 250. Ψ sites detected at chr3:36685288, chr20:37975232, and chr22:42565082 originate from 7SK pseudogenes on different chromosomes. Each of these sites also contain the UGUAG motif at position 250 and exhibited reduced Ψ fractions following PUS7 depletion (Fig. S2F).

### PUS7 binds to 7SK RNA and regulates its pseudouridylation

Previous studies in yeast have demonstrated that PUS7 is responsible for the pseudouridylation of tRNA and U2 snRNA[34]. More recently, PUS7 has been identified as a critical regulator of Ψ-derived tRNA fragmentations, which modulate translation in stem cells[8–10]. To further confirm the RNA substrates of PUS7 in CRC cells, we performed photoactivatable ribonucleoside-enhanced crosslinking and immunoprecipitation (PAR-CLIP) of PUS7 in HCT116 cells to capture potential binding targets. Two replicate PAR-CLIP experiments were highly consistent with each other (Fig. 2A). The overlapped 185 targets were enriched in tRNA and snRNA family (Fig. 2B, Supplementary Data 2), consistent with previously reported PUS7-iCLIP-seq data[8]. When we overlapped these identified targets with the 25 PUS7-dependent Ψ sites, 7SK stood out as a new target of PUS7 (Fig. 2C). The read coverage of PUS7-PAR-CLIP showed that PUS7 preferentially binds to 3'-end of 7SK, instead of the exact position of Ψ site (Fig. 2D), suggesting that human PUS7 recognizes and binds to an extended RNA sequence[35]. The highly structured nature of 7SK may allow PUS7 to recognize its 3' end while still positioning the catalytic site in spatial proximity to the U250 site of 7SK.

7SK is an abundant nuclear ncRNA that mediates the functions of the canonical 7SK ribonucleoprotein (RNP) and barrier-to-autointegration factor (BAF) protein complex[24,25,36]. It contains a pseudouridine (Ψ) modification at position U250, which has been reported to be catalyzed by DKC1[26]. After *PUS7* knockdown, we examined DKC1 expression level and observed no significant change at either the RNA or protein level (Fig. S3A, B). We also performed a combined knockdown of *DKC1* and *PUS7*, and observed that only PUS7 depletion reduced the Ψ level on 7SK in HCT116 cells (Fig. S3I). These findings suggest that DKC1 does not contribute to 7SK pseudouridylation in CRC cells, potentially due to the absence or low abundance of the guiding snoRNA required for DKC1 activity at this locus[26]. This led us to hypothesize that the reduction in Ψ at 7SK was directly mediated by PUS7. To further validate 7SK as a substrate of PUS7, we expressed full-length PUS7 with a His6 tag from *E coli* (Fig. S3C) and transcribed 7SK in vitro (Fig. S3D) for catalytic assay and binding assay. For the catalytic assay, a range of concentration of PUS7 was incubated with 5 pmol of 7SK at 37 °C for 30 min. The resulting Ψ levels were quantified using triple quadrupole LC-MS/MS. The calculated turnover number (TON) of PUS7 towards in-vitro-transcribed 7SK is around 0.3 (Fig. 2E). The dissociation constant ($K_d$) between PUS7 and 7SK was also determined using electrophoretic mobility shift assay (EMSA) and measured as $98 \pm 22$ nM (Fig. 2F).

To evaluate pseudouridylation of 7SK by PUS7 in other CRC cell lines, we conducted bisulfite incorporation hindered ligation-based

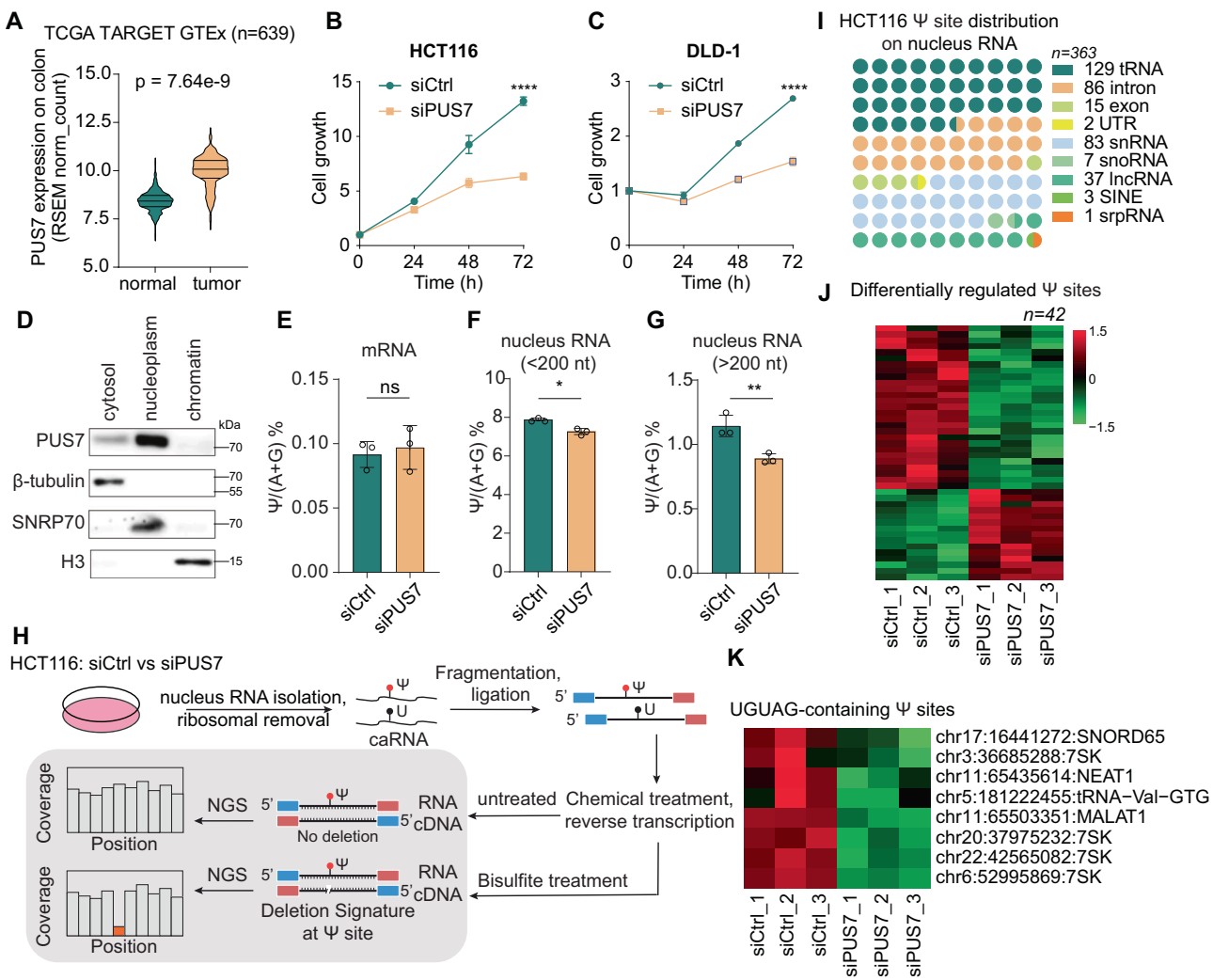

**Fig. 1 | PUS7 regulates pseudouridine (Ψ) levels on nuclear RNA. A** Differential expression of PUS7 in normal and tumor colon tissues from the TCGA GTEx dataset. *p* value was analyzed by two-tailed Student's *t* test. **B** Cell proliferation in the colon cancer cell line HCT116 with or without *PUS7* knockdown. Data are presented as mean ± SEM (*n* = 3, biological replicates), analyzed by two-tailed Student's *t* test. **C** Cell proliferation in the colon cancer cell line DLD-1 with or without *PUS7* knockdown. Data are presented as mean ± SEM (*n* = 3, biological replicates), analyzed by two-tailed Student's *t* test. **D** Western blot analysis of PUS7 in cytosolic, nucleoplasmic, and chromatin fractions, with β-tubulin (cytoplasmic), SNRP70 (nucleoplasmic), and H3 (chromatin) as controls. Cell line used: HCT116. This experiment was repeated independently twice with similar results. **E** Change in Ψ levels on mRNA in HCT116 cells with or without *PUS7* knockdown, quantified by Triple Quad LC-MS/MS. Data are presented as mean ± SEM (*n* = 3, biological replicates), analyzed by two-tailed Student's *t* test. **F** Change in Ψ levels on nucleus RNA (<200 nt, ribosomal RNA-depleted) in HCT116 cells with or without *PUS7* knockdown, quantified by Triple Quad LC-MS/MS. Data are presented as mean ± SEM (*n* = 3, biological replicates), analyzed by two-tailed Student's *t* test. **G** Change in Ψ levels on nuclear RNA (>200 nt, ribosomal RNA-depleted) in HCT116 cells with or without *PUS7* knockdown, quantified by Triple Quad LC-MS/MS. Data are presented as mean ± SEM (*n* = 3, biological replicates), analyzed by two-tailed Student's *t* test. **H** Ψ detection workflow using bisulfite-induced deletion sequencing (BID-seq). **I** Distribution of pseudouridine sites on nuclear RNA in HCT116 cells. **J** Heatmap showing differentially regulated Ψ sites. **K** Heatmap of 8 differentially regulated Ψ sites containing the UGUAG motif, a potential PUS7 substrate motif. *p* value significance: *\*p* < 0.05, *\*\*p* < 0.01, *\*\*\*p* < 0.001, *\*\*\*\*p* < 0.0001; ns not significant. Source data are provided as a Source Data file.

quantitative PCR (BIHIND-qPCR) assay to monitor changes in the Ψ fraction after PUS7 depletion[37]. This method takes advantages of the selective incorporation of bisulfite into Ψ to differentiate Ψ and U (Fig. 2G). We first validated the assay in vitro by testing PUS7's ability to modify 7SK at U250. Incubating in vitro-transcribed 7SK with PUS7-His₆ yielded an 81% Ψ fraction at this site (Fig. S3E). We then applied the assay to cellular systems. After *PUS7* knockdown, the BIHIND-qPCR results revealed that the Ψ level of 7SK Ψ250 in HCT116 was reduced from 72 to 48% (Fig. 2H), consistent with the BID-seq results. Ψ fraction change of 7SK for another colon cancer cell line DLD-1 had also shown the similar trend after *PUS7* KD (Fig. 2I). Additionally, Ψ fraction of 7SK in a cervical cancer cell line HeLa was reduced by 34% (Fig. 2J) after *PUS7* KD, indicating that 7SK Ψ250 is a conservative Ψ site installed by PUS7 in different cancer types.

## Loss of PUS7 promotes release of P-TEFb components from 7SK snRNP

RNA modifications on 7SK are critical for the formation of its secondary structure and association of P-TEFb with 7SK[26–28]. Both Ψ and m⁶A modifications have been reported to influence the stability of 7SK snRNP[26–28]. The m⁶A modification on 7SK is thought to disrupt the secondary structure of 7SK and promotes P-TEFb release from 7SK in A549 cells[28]. To rule out the secondary effect caused by m⁶A changes on 7SK after *PUS7* KD, we performed m⁶A-immunoprecipitation followed by qPCR (MeRIP-qPCR) on 7SK. Interestingly, the m⁶A level by MeRIP-qPCR on 7SK is very low in CRC cells (Fig. S3F, left) and no significant m⁶A level change was observed after *PUS7* knockdown (Fig. S3F, right). Pseudouridylation contributes to RNA folding due to the extra hydrogen bond donor on the Ψ base, which can organize a

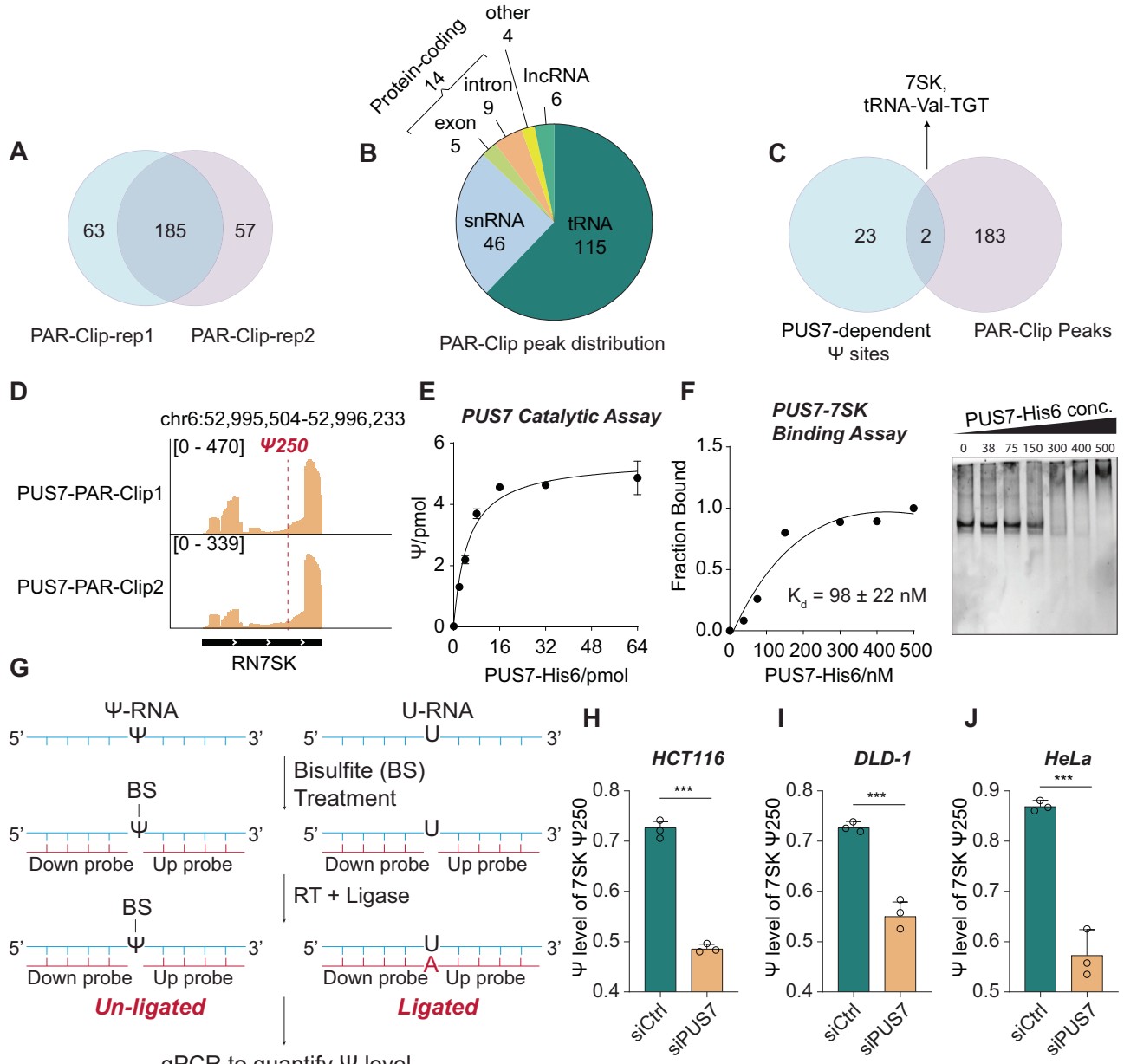

**Fig. 2 | 7SK is a conserved substrate of PUS7. A** Peak overlap between two replicates of PUS7-PAR-CLIP. **B** Pie chart showing the distribution of RNA species bound by PUS7, identified by PUS7-PAR-CLIP in HCT116 cells. **C** Pie chart showing the overlap between PUS7-dependent Ψ sites and peaks identified by PUS7-PAR-CLIP. **D** Integrative Genomics Viewer (IGV) plots showing read coverage of PUS7-PAR-CLIP peaks on 7SK RNA, with the Y-axis indicating normalized counts. **E** In vitro catalytic assay showing PUS7 activity towards in vitro-transcribed 7SK. The X-axis represents the amount of PUS7 used, and the Y-axis shows the Ψ level produced. Data are presented as mean ± SEM ($n = 2$, biological replicates). **F** Gel-shift assay showing the binding affinity between PUS7 and in vitro-transcribed 7SK. Left panel: binding curve between PUS7 and 7SK; right panel: gel shift data. Data are presented as mean ± SEM ($n = 2$, biological replicates). **G** Workflow of BIHIND-qPCR for detecting and quantifying Ψ levels at specific sites. **H** Validation of Ψ level changes on 7SK in HCT116 cells after *PUS7* knockdown by BIHIND-qPCR. Data are presented as mean ± SEM ($n = 3$, biological replicates), analyzed by two-tailed Student's *t* test. **I** Bar plot showing Ψ level changes on 7SK in DLD-1 cells after *PUS7* knockdown by BIHIND-qPCR. Data are presented as mean ± SEM ($n = 3$, biological replicates), analyzed by two-tailed Student's *t* test. **J** Bar plot showing Ψ level changes on 7SK in HeLa cells after *PUS7* knockdown by BIHIND-qPCR. Data are presented as mean ± SEM ($n = 3$, biological replicates), analyzed by two-tailed Student's *t* test. *p* value significance: *$p < 0.05$, **$p < 0.01$, ***$p < 0.001$, ****$p < 0.0001$; ns not significant. Source data are provided as a Source Data file.

water molecule[38]. Since depletion of PUS7 led to decreased pseudouridylation at 7SK Ψ250, we expected that more P-TEFb complexes would be released from the 7SK RNA (Fig. 3A). To test this, we immunopurified 7SK snRNP complexes from lysates of *PUS7* KD and control cells using antibodies against LARP7, HEIMX1 or MEPCE and monitored co-precipitation of CDK9 and Cyclin T1 (Fig. 3B). Notably, we observed substantially reduced amounts of CDK9 and cyclin T1 co-immunoprecipitated with anti-LARP7, anti-HEXIM1 or anti-MEPCE in *PUS7* KD cells (Fig. 3C). Specifically, the relative binding of Cyclin T1 to

LARP7, HEXIM1, and MEPCE was reduced by 67%, 39%, and 71%, respectively, in *PUS7* KD cells compared to controls. Similarly, the relative binding of CDK9 to LARP7, HEXIM1, and MEPCE was reduced by 57%, 35%, and 47%, respectively. This result supports our hypothesis that the association between P-TEFb and 7SK is weakened and P-TEFb is released from 7SK snRNP upon *PUS7* KD. Heterogeneous nuclear ribonucleoproteins (hnRNPs), including hnRNP A1, A2/B1, Q, and R can associate with 7SK after P-TEFb releasing from 7SK[39–41]. Consistent with this model, we observed a significant increase in the association of

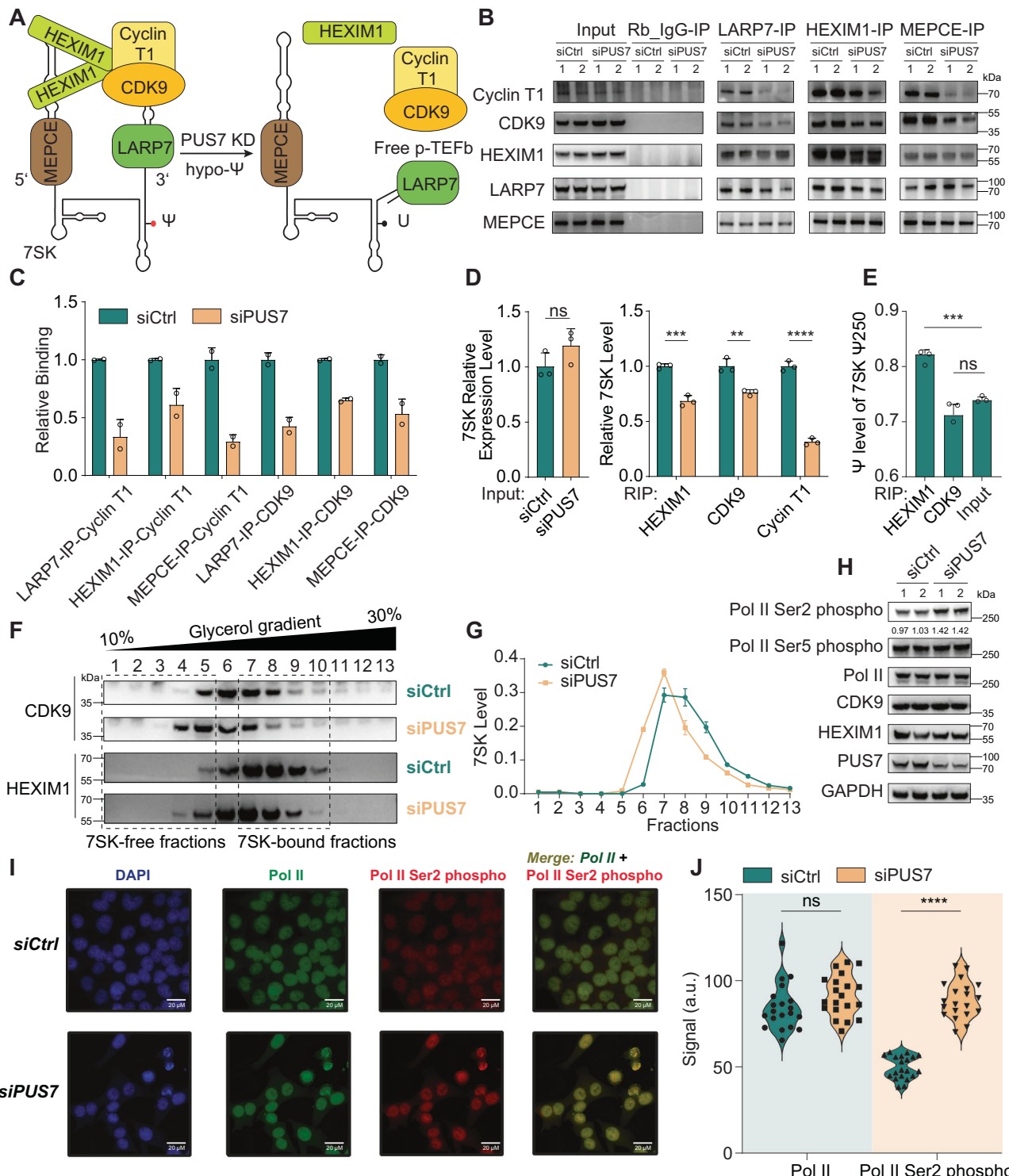

hnRNP A1, Q, and R with 7SK (pulled down via anti-LARP7), despite no changes in their overall expression levels (Fig. S3G).

We further pulled down the 7SK complex by immunoprecipitation of HEXIM1, CDK9 or cyclin T1, and then measured the 7SK RNA level by qPCR. A lower immunoprecipitated 7SK level was observed after *PUS7* KD although the whole 7SK level remained largely unchanged (Fig. 3D). Specifically, the relative 7SK RNA level co-immunoprecipitated with HEXIM1 was reduced by 31%, with CDK9 by 24%, and with Cyclin T1 by 69% in *PUS7* KD cells compared to controls. Additionally, we assessed the Ψ level of 7SK immunoprecipitated by anti-HEXIM1 or anti-CDK9 by BIHIND-qPCR. Higher Ψ fraction was observed on 7SK pulled down

with HEXIM1 compared to input but not with CDK9 (Fig. 3E). This finding suggests that HEXIM1 may preferentially associate with pseudouridylated 7SK to anchor P-TEFb complex, consistent with previous observations[26,42]. The reason why CDK9-immunoprecipation doesn't enrich pseudouridylated 7SK likely reflects the requirement of 7SK–HEXIM1 interaction to associate P-TEFb[42]. Pseudouridylation at Ψ250 primarily facilitates 7SK–HEXIM1 binding, which is upstream of CDK9 recruitment. Upon *PUS7* knockdown, the reduced 7SK pseudouridylation weakens its interaction with HEXIM1, leading to dissociation of the complex and subsequent release of P-TEFb. Therefore, the effects of pseudouridylation are primarily mediated through

**Fig. 3 | Loss of PUS7 promotes the release of P-TEFb components from the 7SK snRNP. A** Schematic representation of PUS7 regulation on the stability of the 7SK snRNP complex. **B** Western blot analysis of Cyclin T1, CDK9, HEXIM1, LARP7, and MEPCE immunoprecipitated with antibodies against LARP7, HEXIM1, or MEPCE. **C** Quantification of CDK9 and Cyclin T1 levels co-immunoprecipitated with LARP7, HEXIM1, or MEPCE. Data are shown as mean ± SEM ($n = 2$). **D** Bar plots showing 7SK RNA levels immunoprecipitated by antibodies against HEXIM1, CDK9, or Cyclin T1, with or without *PUS7* knockdown. Left panel: input 7SK levels; right panel: immunoprecipitated 7SK levels. Data are presented as mean ± SEM ($n = 3$, biological replicates) and analyzed using two-tailed Student's *t* tests. **E** Ψ fraction of 7SK Ψ250 in complexes immunoprecipitated by HEXIM1 or CDK9, determined by BIHIND-qPCR. Data are shown as mean ± SEM ($n = 3$, biological replicates) with two-tailed Student's *t* tests. **F** Glycerol gradient sedimentation (10–30%) of HCT116 lysates followed by immunoblotting for CDK9 and HEXIM1. Two regions "7SK-free fractions" and "7SK-bound fractions" were defined based on 7SK level in each

fraction. The gel shown was representative of two replicates. **G** Distribution of 7SK RNA across glycerol gradient fractions (10–30%), quantified by qPCR. Data are presented as mean ± SEM ($n = 2$, biological replicates). **H** Western blot showing changes in Pol II phosphorylation levels with or without PUS7 *knockdown*. This experiment was repeated independently twice with similar results.
**I** Immunostaining showing that PUS7 depletion increases RNA Polymerase II Serine 2 phosphorylation. Left: Immunofluorescence images of DAPI (nuclei), RNA Polymerase II, and Pol II Serine 2 phosphorylation with or without *PUS7* KD. Merged images of Pol II and Pol II Ser2 phospho are also shown. This experiment was repeated independently twice with similar results. **J** Violin plots quantifying Pol II Ser2 phosphorylation intensity in siCtrl and siPUS7 cells. Each point represents a single cell. Data is analyzed using two-tailed Student's *t* tests. *p* value significance: \**p* < 0.05, \*\**p* < 0.01, \*\*\**p* < 0.001, \*\*\*\**p* < 0.0001; ns, not significant. Source data are provided as a Source Data file.

HEXIM1, and the phenotype observed in CDK9 pulldown is a secondary mitigated consequence.

We then performed a glycerol gradient assay to monitor the size distribution of the 7SK RNP (Fig. 3F) and quantified 7SK levels in each fraction (Fig. 3G). Based on the relative abundance of 7SK in each fraction, we defined fractions 1–5 as 7SK-free and fractions 7–10 as 7SK-bound. Following *PUS7* knockdown, HEXIM1 and CDK9 were observed to shift from 7SK-bound fractions, which correspond to the larger-sized 7SK snRNP, to 7SK-free fractions, which represent the smaller-sized 7SK snRNP (Fig. 3F). 7SK also shifted towards the gradients associated with smaller-size 7SK snRNP (Fig. 3G), confirming the enhanced release of P-TEFb from 7SK RNP after *PUS7* KD. The release of P-TEFb from 7SK RNP is supposed to drive the phosphorylation of serine 2 (Ser2) on the carboxy terminal domain (CTD) of Pol II[20,21]. We indeed observed that Ser2 phosphorylation (S2P) level was notably increased by 40% with PUS7 depletion, while Ser5 phosphorylation (S5P) was not affected (Fig. 3H). This increase in Ser2 phosphorylation was independently confirmed by microscopy-based quantification (Fig. 3I, J).

7SK is reported to interact with the BAF chromatin-remodeling complex, playing a role in regulating chromatin remodeling at enhancers[36]. Following a glycerol gradient assay to assess the size distribution of the BAF complex, we found that PUS7 depletion did not noticeably affect the size distribution of the BAF complex (Fig. S3H).

## PUS7 depletion promotes transcription elongation

P-TEFb plays a crucial role in transcription regulation by facilitating the release of Pol II from promoter-proximal pausing through the phosphorylation of Pol II CTD Ser2[18–21,43]. Having identified the increased phosphorylation of Pol II CTD Ser2 with PUS7 depletion, we next asked whether transcription elongation could be affected. Here, we used 7SK antisense oligonucleotides (7SK ASO) to deplete 7SK transiently as the positive control (Fig. S4A). To gain more mechanistic insights into Ψ −7SK-related transcription regulation, we performed kethoxal-assisted single-stranded DNA sequencing (KAS-seq) to monitor the transcription process in *PUS7* KD, 7SK ASO, and control cells. KAS-seq specifically captures and measures single-stranded DNA (ssDNA), which reveals the dynamics of transcriptionally engaged Pol II, allowing for highly sensitive genome-wide profiling of active transcription[44,45]. Accordingly, KAS-seq signals are highly correlated with the signals of Pol II chromatin immunoprecipitation sequencing (ChIP-seq)[44,45]. Metagene analysis of the KAS-seq data showed that both reducing Ψ level of 7SK Ψ250 by *PUS7* KD and depleting 7SK with 7SK-ASO resulted in accumulation of Pol II in gene body regions and reduction around promoter regions (Fig. 4A). Heatmaps ranked by decreasing KAS-seq signals confirmed the enhanced Pol II elongation by *PUS7* KD (Fig. S4B), consistent with the metagene analysis.

We estimated the ratio of Pol II occupancy between the gene body and its promoter as "Pol II release ratio" (PRR)[46]. Specifically, the

promoter region is defined as 100 bp upstream and 300 bp downstream of the TSS, and the gene body region is defined as the region from 300 bp to 2 kb downstream of the TSS (Fig. 4B). The KAS-seq data is highly consistent between each pair of replicates (Fig. S4C–S4E, Supplementary Data. 3). 7SK depletion by 7SK ASO resulted in the release of P-TEFb and promoted the transcription elongation. The log$_2$[PRR] values of the 2058 detected active genes were significantly increased after 7SK loss, confirming enhanced transcription elongation (Fig. 4C). A similar trend was observed in *PUS7* KD cells, further indicating that PUS7 depletion impairs 7SK binding to the P-TEFb complexes (Fig. 4C).

We next selected three representative genes (*EGR1*, *DDIT3* and *KLF6*). Both *PUS7* KD and 7SK ASO treatments led to increased read coverage of ssDNA within gene bodies (Fig. 4D–F). We validated these results in a Pol II-ChIP-qPCR (chromatin immunoprecipitation of Pol II followed by qPCR) approach (Fig. 4G). This enhanced transcription elongation also correlated with increased gene expression in the mRNA level (Fig. 4H). When we performed the gene ontology (GO) enrichment analysis with the top 1000 PRR-increased genes, we noticed that the apoptotic pathway and genes involved in regulation of apoptosis were both highlighted in *PUS7* KD and 7SK ASO group (Fig. 4I, J). Specifically, within the apoptosis-related gene set, 49 genes contributed to these enrichments in the *PUS7* KD group, while 47 genes contributed in the 7SK ASO group (from a total apoptosis gene set of 593 genes).

## PUS7- and 7SK-deficient cells share partially overlapping transcriptome alterations

The KAS-seq data indicates that both *PUS7* KD and 7SK ASO cells exhibit enhanced transcription elongation through the releasing of paused P-TEFb. We next extracted mRNA from *PUS7* KD, 7SK ASO, and control cells, and performed RNA-seq. Principal component analysis (PCA) revealed that *PUS7* KD and 7SK ASO samples cluster closely together, while both are distant from the control samples along PC1, which accounts for 50% of the variance (Fig. S5A). We conducted differential expression analysis to quantify transcriptome alterations and identified 1,089 significantly changed transcripts (606 upregulated and 483 downregulated) in *PUS7* KD cells compared to the control and 769 significantly changed transcripts (440 upregulated and 329 downregulated) in 7SK ASO cells compared to the control (Fig. 4K, L, Supplementary Data. 4). Among these differentially expressed transcripts, 205 upregulated genes and 149 downregulated genes were shared between *PUS7* KD and 7SK ASO cells (Fig. 4M). The correlation heatmap also illustrates that the gene expression profiles were highly correlated between PUS7- and 7SK-deficient cells (Fig. 4N). To correlate the enhanced transcription elongation with the mRNA alterations, we combined the KAS-seq data and mRNA-seq data and found that those genes with increased PRR values tend to exhibit higher positive changes of their mRNA expression levels (Fig. S5B, C).

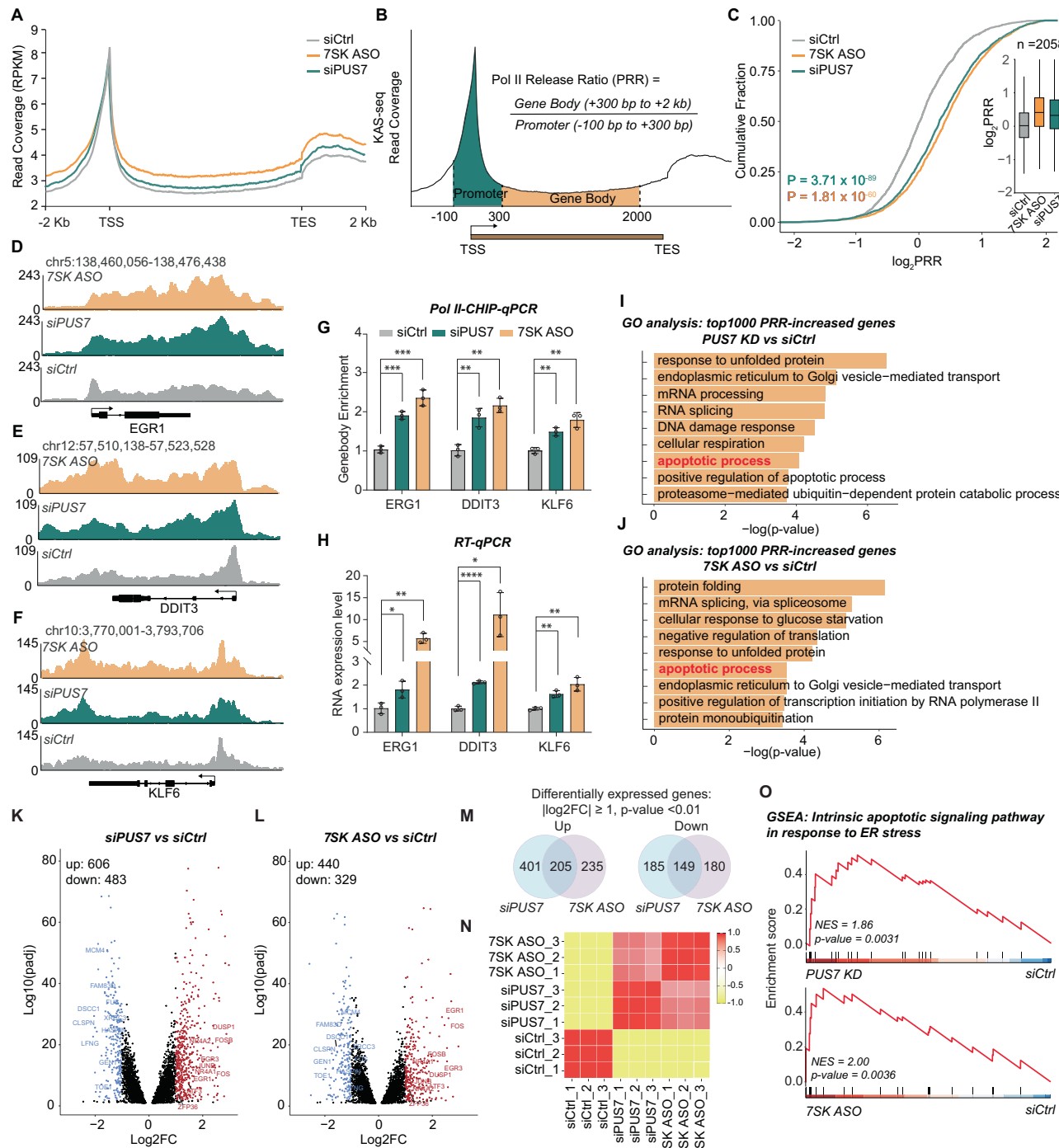

**Fig. 4 | PUS7 or 7SK depletion enhances transcription elongation. A** Metagene analysis displaying KAS-seq read coverage across 7962 genes in HCT116 cells. Read coverage was displayed with RPKM (Reads Per Kilobase per Million). **B** Schematic representation of the gene body/promoter ratio used to calculate the Pol II release ratio (PRR). **C** Cumulative distribution and boxplots (inset) of $\log_2$PRR. Box plots show the median (center line), 25th–75th percentiles (box), and minima/maxima (whiskers). Statistical significance was determined by two-sided Wilcoxon rank-sum test. Integrative Genomics Viewer (IGV) plots illustrating KAS-seq read coverage for EGR1 (**D**), DDIT3 (**E**), and KLF6 (**F**) in HCT116 cells. Y-axis indicates normalized read counts. **G** Bar plot showing enrichment of Pol II in gene body over the promoter in HCT116 cells with PUS7 or 7SK depletion, determined using Pol II ChIP-qPCR. Data are presented as mean ± SEM ($n = 3$, biological replicates) and analyzed using two-tailed Student's $t$ tests. **H** Bar plot showing RNA expression changes of *EGR1*, *DDIT3*, and *KLF6* in HCT116 cells with PUS7 or 7SK depletion. Data are presented as mean ± SEM ($n = 3$, biological

replicates) and analyzed using two-tailed Student's $t$ tests. Gene ontology (GO) analysis of the top 1000 genes with increased PRR after *PUS7* knockdown (**I**) or 7SK depletion (**J**). $p$ values were calculated using a hypergeometric test without adjustment for multiple comparisons. Volcano plots showing adjusted p-values (padj) and fold changes in transcript levels from mRNA-seq with or without *PUS7* knockdown (**K**) or 7SK depletion (**L**). Cutoffs: $|\log_2\text{FC}| \geq 1$, adjusted $p < 0.01$ using DESeq2 with Benjamini–Hochberg correction. **M** Venn diagram of altered transcripts in PUS7- and 7SK-depleted cells. **N** Correlation heatmap of transcriptome profiles among PUS7-depleted, 7SK-depleted, and control cells. **O** Gene set enrichment analysis (GSEA) highlighting the upregulation of the intrinsic apoptotic signaling pathway. NES: normalized enrichment score. Statistical significance was assessed using a two-sided permutation test implemented in the R package clusterProfiler, and reported $p$ values are unadjusted. $p$ value significance: \*$p < 0.05$, \*\*$p < 0.01$, \*\*\*$p < 0.001$, \*\*\*\*$p < 0.0001$; ns not significant. Source data are provided as a Source Data file.

To identify the downstream signaling pathway shared in PUS7- and 7SK-deficient cells compared to the control cells, we performed GO analysis and discovered that both PUS7 and 7SK depletion stimulated apoptotic process (Fig. S5D, S5E), aligning well with the KAS-seq results (Fig. 4I, J). Specially, gene set enrichment analysis (GSEA) showed significant upregulation of intrinsic apoptotic signaling pathway in response to ER stress in *PUS7* KD and 7SK ASO cells compared to control cells (Fig. 4O). Reanalysis of a public study on GBM stem cell with *PUS7* KD also indicated significant upregulation of ER stress-mediated apoptosis (Fig. S5F, GSE147382). A comparison of gene expression profiles between GBM stem cells and CRC cells upon *PUS7* KD revealed a notable overlap between differentially expressed genes (Fig. S5G).

## KLF6/DDIT3 is involved in PUS7/Ψ/7SK downstream pathway

KAS-seq and mRNA-seq analyses revealed that PUS7 depletion results in hypo-pseudouridylation of 7SK, enhanced Pol II transcription elongation, and increased expression of KLF6 and DDIT3 (Fig. 5A, B). As a tumor suppressor, KLF6 is frequently inactivated in colon cancer[47], with low KLF6 expression associated with worse overall survival as shown by Kaplan–Meier analysis of the TCGA-COAD dataset (Fig. S6). Given that KLF6 is a zinc finger transcription factor known to bind and activate ATF3 expression[48], we hypothesize that reactivating KLF6 expression through PUS7 depletion may drive ATF3/DDIT3-mediated apoptosis in CRC cells (Fig. 5A). To test this, we performed transient knockdown of *KLF6* or *DDIT3* alongside PUS7 depletion and observed rescued cell viability (Fig. 5C). The knockdown of either *KLF6* or *DDIT3* also reduced apoptosis ratios caused by PUS7 depletion, underscoring the critical roles of KLF6 and DDIT3 in mediating the apoptotic effects induced by PUS7 depletion (Fig. 5C–E). Gene expression correlation analysis confirmed a strong inverse relationship between *PUS7* expression and *KLF6* expression, further supporting the regulatory link between these genes (Fig. 5F). We also observed a positive correlation between *PUS7* expression and *DDIT3* expression (Fig. S7). This apparent discrepancy may reflect the fact that DDIT3 is regulated by multiple upstream factors, including activating transcription factors (ATFs) and X-box binding protein 1 (XBP1)[49]. The involvement of these additional pathways may obscure or inverse the correlation between *PUS7* and *DDIT3*.

The fluorinated analog of uracil, 5-fluorouracil (5-FU), is a cornerstone chemotherapeutic agent for both palliative and adjuvant treatment of colorectal cancer (CRC)[50,51]. Previous studies have shown that increased expression of DDIT3 sensitizes CRC cells to 5-FU[52]. Since PUS7 depletion upregulates DDIT3 expression via the PUS7/Ψ/7SK axis, we hypothesized that *PUS7* KD might similarly enhance CRC cell sensitivity to 5-FU. We treated HCT116 and HCT-15 CRC cells with a range of 5-FU concentrations following PUS7 or 7SK depletion and assessed cell viability. The results showed that both cell lines exhibited increased sensitivity to 5-FU, as reflected by a leftward shift in their dose–response curves (Fig. 5G). Furthermore, overexpression (OE) of wild-type PUS7 (PUS7 WT) desensitized CRC cells to 5-FU with survival maintained at higher drug concentrations, while overexpression of a catalytically inactive PUS7 mutant (D294N) had no such effect (Fig. 5H). To explore potential therapeutic implications, we employed a small-molecule PUS7 inhibitor[10], NSC107512, which reduced HCT116 cell viability by approximately 40% at 0.125 μM (Fig. S8A), mirroring the effects of *PUS7* knockdown. At this concentration, NSC107512 significantly enhanced 5-FU sensitivity (Fig. S8B), supporting the feasibility of pharmacological PUS7 inhibition as a strategy to potentiate 5-FU efficacy in CRC.

To further confirm that pseudouridylation of 7SK Ψ250 is the key regulator of PUS7 downstream pathway in CRC, we employed a catalytically inactive RNA-targeting CRISPR-Cas13 system to specifically manipulate pseudouridylation of 7SK. By fusing the dCas13b system with PUS7, we tested whether these tethering constructs could modulate downstream gene expression (Fig. 5I). Tethering PUS7 increased the pseudouridylation level of 7SK Ψ250, as verified by BIHIND-qPCR (Fig. 5J), reduced the phosphorylation level of Pol II Ser2, and downregulated the expression levels of KLF6 and DDIT3 (Fig. 5L), collectively leading to enhanced cell proliferation (Fig. 5K). Moreover, targeting pseudouridylation of 7SK with dCas13b-PUS7 significantly conferred stronger resistance to 5-FU in CRC cells (Fig. 5M). To rule out the possibility that dcas13b-PUS7 binding to 7SK causes the observed phenotype, we generated dcas13b without PUS7 conjugation (Fig. S9A) and conducted the same experiments. The previously observed effects—including increased Ψ at 7SK U250, altered proliferation, changes in downstream targets (Pol II S2P, KLF6, and DDIT3), and sensitivity to 5-FU—were no longer observed (Fig. S9B–S9E). These findings reveal that PUS7 negatively regulates the KLF6/DDIT3-mediated apoptosis through the PUS7/Ψ/7SK axis, influencing CRC cell sensitivity to chemotherapeutic treatment.

## Discussion

Pseudouridine (Ψ) is one of the highly abundant RNA modifications in mammalian cells, and plays important roles in several RNA types, including mRNA, tRNA, snRNA, and rRNA[5–7]. It provides an extra hydrogen bonding donor to RNA base, and can influence a range of different biological processes involving RNA. Ψ within pre-mRNA intronic regions or spliceosomal snRNAs can impact splicing[15,34]. In translation, Ψ is critical for maintaining accuracy in rRNA[12,53,54], and regulates tRNA function through Ψ-derived tRNA fragmentation[7–9]. Ψ levels in rRNA and tRNA correlate with translation activity in plants[55]. Recent research has also suggested that Ψ on mRNA may play a role on mRNA stability[13]. Abnormal pseudouridylation may disrupt normal life processes and cause human diseases.

Recent advancements in single-base resolution sequencing methodologies have enabled the study of pseudouridylation (Ψ) across the whole transcriptome[12,13], offering enabling tools to dissect functions of individual Ψ sites. In this study, we utilized BID-seq to identify PUS7 as the writer of 7SK Ψ250 in CRC cells and validated its role through a series of in vitro biochemical assays. Dyskerin (DKC1) has also been reported to contribute to the pseudouridylation of 7SK Ψ250 with the assistance of H/ACA box snoRNA[26], indicating that both PUS7 and DKC1 can co-regulate the pseudouridylation of 7SK Ψ250. As a highly abundant snRNA, 7SK plays a very important role in tuning the function of P-TEFb complex and BAF complex and regulating transcription process in mammalian cells[24,25,36]. Our findings show that 7SK Ψ250 restricts the activity of the P-TEFb complex (Fig. 6). Depletion of PUS7 results in hypo-pseudouridylation of 7SK, leading to the dissociation of P-TEFb from 7SK snRNP (Fig. 6, lower panel). This dissociation facilitates Pol II release from promoter-proximal pausing and enhances transcription elongation. The increased transcription of the downstream *KLF6* and *DDIT3* modulates apoptosis, which impacts resistance of CRC to 5-FU chemotherapy. To the best of our knowledge, this study uncovers a new role for Ψ in gene regulation at the transcriptional level. Our research identified 7SK as a key downstream substrate of PUS7, highlighting the potential of targeting 7SK-mediated pathways through PUS7 inhibition. New possibilities in anti-cancer therapies may emerge from these discoveries. However, a limitation of our study is the partial (~50%) knockdown efficiency of *PUS7*, which may underestimate the extent of its phenotypic impact, including the Ψ changes on mRNA.

$N^6$-methyladenosine (m$^6$A) modification plays crucial roles in gene expression regulation[1–3]. In non-small cell lung cancer cells, 7SK is found to be highly modified with m$^6$A, with methyltransferase-like 3 (METTL3) and alkB homolog 5 (ALKBH5) identified as the responsible writer and eraser[28]. The presence of m$^6$A induces a structural change in 7SK that promotes the sequestration of P-TEFb, leading to decreased phosphorylation of Pol II. It is interesting that both Ψ and m$^6$A modifications on 7SK are involved in transcription regulation. These modifications may respond to different cell signaling and/or

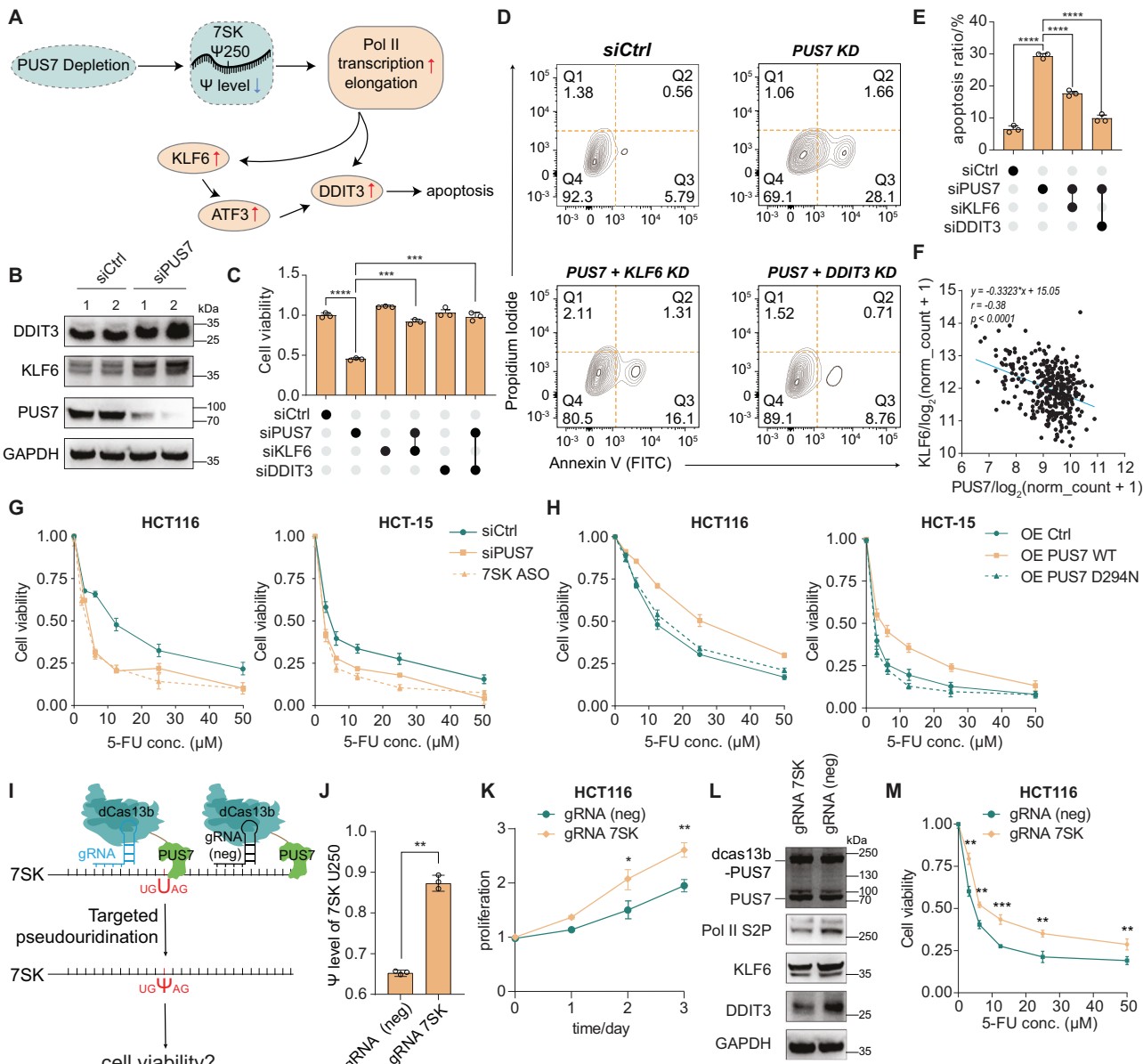

**Fig. 5 | PUS7 depletion induces KLF6/DDIT3-mediated apoptosis and sensitizes colon cancer cells to 5-FU. A** Proposed model of the downstream pathway of PUS7 depletion. **B** Western blot showing DDIT3 and KLF6 upregulation after *PUS7* knockdown. This experiment was repeated independently twice with similar results. **C** Rescue of cell viability by co-knockdown of *DDIT3* or *KLF6* with *PUS7* in HCT116. Data are presented as mean ± SEM (*n* = 3, biological replicates) with two-tailed Student's *t* tests. **D**, **E** Flow cytometry analysis of apoptosis using Annexin V and propidium iodide staining in HCT116. Data are shown as mean ± SEM (*n* = 3, biological replicates) with two-tailed Student's *t* tests. **F** Correlation analysis between PUS7 and KLF6 expression in the TCGA COAD dataset. Pearson's correlation coefficient (r) and associated *p* value (two-tailed test) are shown. **G** Dose-response curves for HCT116 and HCT-15 cells treated with 5-FU after siCtrl, siPUS7, or 7SK ASO treatment. Data are presented as mean ± SEM (*n* = 3, biological replicates). **H** Dose-response curves for HCT116 and HCT-15 cells with overexpression

(OE) of control, wild-type (WT) PUS7, or mutant PUS7 (D249N). Data are presented as mean ± SEM (*n* = 3, biological replicates). **I** Schematic diagram of the dCas13b system fused with PUS7 (dCas13b-PUS7). **J** Bar plot showing the Ψ level at 7SK Ψ250 upon introducing the dCas13b-PUS7 system with or without guide RNA targeting 7SK U250. Data are presented as mean ± SEM (*n* = 3, biological replicates) with two-tailed Student's *t* tests. **K** Cell proliferation in HCT116 cells using the dCas13b-PUS7 system. Data are shown as mean ± SEM (*n* = 3, biological replicates) with two-tailed Student's *t* tests. **L** Western blot of Pol II S2p, KLF6, and DDIT3 levels in HCT116 cells using the dCas13b-PUS7 system. This experiment was repeated independently twice with similar results. **M** Dose-response curves of HCT116 cells treated with 5-FU using the dCas13b-PUS7 system. Data are presented as mean ± SEM (*n* = 3, biological replicates) and analyzed using two-tailed Student's *t* tests. *p* value significance: \**p* < 0.05, \*\**p* < 0.01, \*\*\**p* < 0.001, \*\*\*\**p* < 0.0001; ns not significant. Source data are provided as a Source Data file.

exert effects in different cellular contexts. For instance, the m⁶A levels may differ significantly across various cancer types, influencing the extent to which m⁶A effects on 7SK contribute to transcription elongation. Exploring the potential crosstalk between m⁶A and Ψ modifications on 7SK in the future, particularly across diverse biological settings, will be highly valuable for a complete understanding of Pol II activity regulation.

## Methods

### Cell culture

Human HCT116, DLD-1, HCT15, and HeLa cell lines used in this study were all purchased from ATCC (the American Type Culture Collection). HCT116, DLD-1, and HCT15 were grown in RPMI (Gibco, 22400) supplemented with 10% FBS and 1% 100x Pen/Strep (Gibco). HeLa cell line was grown in DMEM (Gibco, 11995) media supplemented with 10% FBS

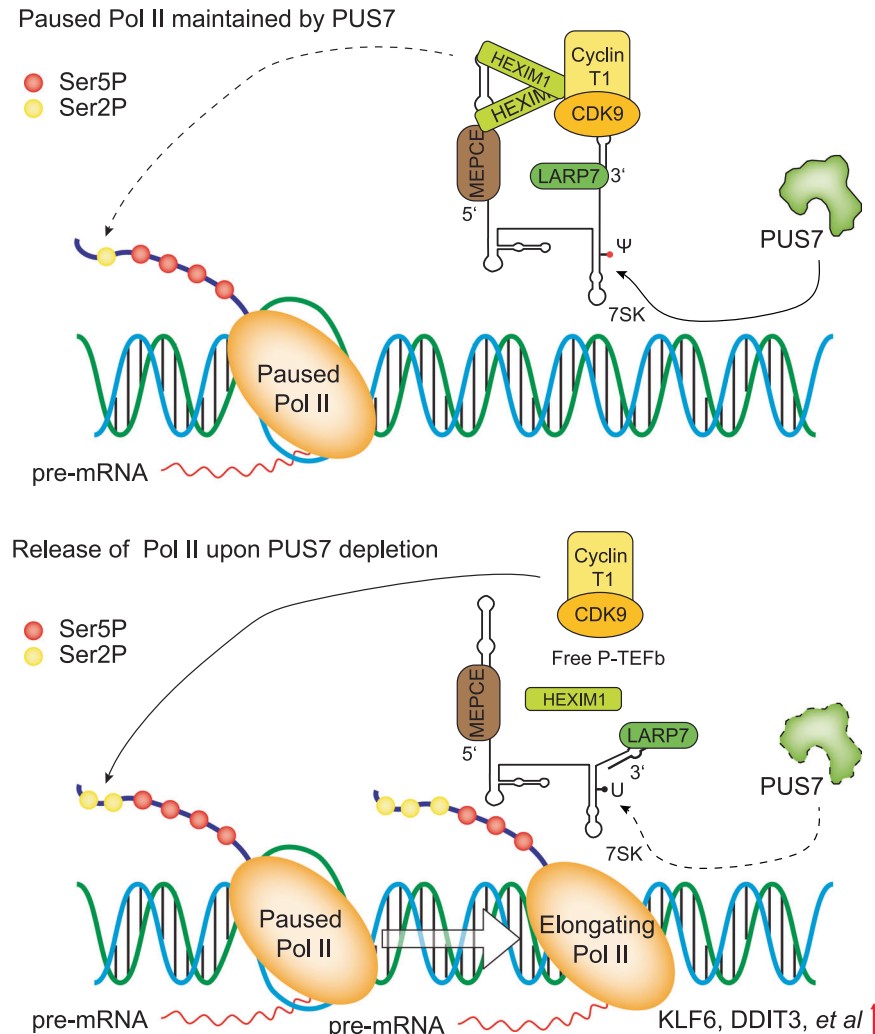

**Fig. 6 | A schematic model of PUS7 in transcription elongation regulation.** PUS7 installs 7SK Ψ250. PUS7 depletion results in hypo-peseudouridylation of 7SK and facilitates the dissociation of P-TEFb from 7SK. The released P-TEFb enhances transcription elongation by increasing phosphorylation of serine 2 (Ser2P) on carboxy terminal domain of RNA Polymerase II, thereby promoting expression of *KLF6* and *DDIT3*.

and 1% 100x Pen/Strep (Gibco). All cells were cultured at 37 °C under 5.0% $CO_2$.

## Antibodies

The antibodies used in this study are listed below in the format of name (supplier, catalog): Rabbit anti-β-Tubulin (Cell Signaling Technology, 2128); Rabbit anti-SNRP70 (abcam, ab83306); Rabbit anti-H3 (Cell Signaling Technology, 9715); Rabbit anti-PUS7 for PAR-CLIP and western blot (Bethyl Laboratories, A305-146A-T (thermo fisher)); Rabbit anti-GAPDH (Cell Signaling Technology, 2118); Rabbit anti-DKC1 (Cell Signaling Technology, 53234); Rabbit anti-LARP7 (Proteintech, 17067-1-AP); Rabbit anti-MEPCE (Proteintech, 14917-1-AP); Rabbit anti-HEXIM1 for immunoprecipitation (Cell Signaling Technology, 12604); Mouse anti-HEXIM1 for western blot (Proteintech, 66311-1-Ig); Rabbit anti-CDK9 (Proteintech, 11705-1-AP); Rabbit anti-ARID1A (Cell Signaling Technology, 12354); Rabbit anti-ARID2 (Cell Signaling Technology, 82342); Rabbit anti-CDK9 (Cell Signaling Technology, 53234); Mouse anti-Pol II (Biolegend, 664906); Rabbit Anti-KLF6 (Invitrogen, PA5-87561(thermo fisher)); Mouse anti-DDIT3 (Cell Signaling Technology, 2895); Goat anti-rabbit IgG-HRP (Cell Signaling Technology, 7074); Goat Anti-Mouse IgG H&L (Alexa Fluor® 488, abcam, ab150113); Goat Anti-Rabbit IgG H&L (Alexa Fluor® 594, abcam, ab150080); Horse anti-mouse IgG-HRP (Cell Signaling Technology, 7076).

## Western blot

Cells were lysed in RIPA Buffer (Thermo Scientific) containing 1x protease inhibitor cocktail (Roche) on ice for at least 15 min. The lysates were centrifuged to remove cellular debris, boiled at 95 °C with 4× loading buffer (Bio-Rad) for 5 min. A total of 20 μg or more protein per sample was loaded into a 4–12% NuPAGE Bis-Tris gel (Life Technologies) and transferred to nitrocellulose membranes (Bio-Rad). Membranes were blocked in 5% milk in TBST for 10 min at room temperature, incubated with a diluted primary antibody solution overnight at 4 °C, washed, and then incubated with an HRP-conjugated secondary antibody for 30 min at room temperature. Protein bands were detected using Immobilon Western Chemiluminescent HRP Substrate (Millipore Sigma) and visualized using iBright FL1500 Imaging System (Thermo Scientific).

## siRNA knockdown and plasmid transfection

siRNA knockdown was performed using Lipofectamine RNAiMAX (Invitrogen 13778150) following manufacture's protocol. The final concentration of siRNA was 20 nM. Following siRNA knockdown, cells were cultured for 3 days before harvest for downstream analyses. Plasmid transfection was performed using Lipofectamine™ 3000 Transfection Reagent (Invitrogen L3000001). Following plasmid transfection, cells were cultured for 2 days before harvest for downstream analyses.

## RNA isolation

To isolate total RNA from cells, the cells were collected and washed once with an appropriate volume of ice-cold DPBS buffer. Cells were then lysed using TRIzol reagent (Invitrogen) following the manufacturer's protocol. RNA was extracted by chloroform, followed by ethanol precipitation or RNA Clean & Concentrator Kits (Zymo). For mRNA enrichment, total RNA was subjected to two rounds of poly(A)+ purification using the Dynabeads mRNA DIRECT kit (Ambion). RNA concentration was determined either by measuring UV absorbance at 260 nm or using the Qubit RNA HS Assay Kit (Thermo Fisher Scientific) with a Qubit 2.0 fluorometer.

## Glycerol gradient assay

HCT116 cells from two 15-cm dishes were collected, and nuclei were isolated by suspending the cells in 3 mL of hypotonic buffer (10 mM Tris-HCl, pH 7.5, 10 mM KCl, 1 mM $MgCl_2$, 1 mM DTT) and incubating on ice for 10 min. The nuclei were collected by centrifugation at 4 °C and washed once with DPBS. The isolated nuclei were lysed in 1.5 mL of high-salt buffer (50 mM Tris-HCl, pH 7.5, 300 mM KCl, 1 mM $MgCl_2$, 1 mM DTT, 1 mM PMSF, 1% NP-40), and the lysate was cleared by centrifugation. The nuclear lysate was layered onto a 10–30% glycerol gradient prepared in 20 mM HEPES-KOH, pH 8.0, 100 mM KCl, 10 mM $MgCl_2$, 1 mM DTT, and 1:100 protease inhibitor cocktail (Roche) and centrifuged in an SW 41 Ti rotor (Beckman Coulter) at 38,000 rpm for 16 h at 4 °C.

## dCas13b-PUS7 assay

dCas13b plasmid was a gift from Dr. Bryan Dickson (University of Chicago). dCas13b-PUS7 and was generated accordingly. The plasmids were sequenced by the University of Chicago Comprehensive Cancer Center DNA Sequencing and Genotyping Facility. For 6-well assays, cells were transfected with 1 μg dCas13b-PUS7 and 1.5 μg gRNA for 24 h before further experiment.

## Cell fractionation

HCT116 cells were fractionated according to a modified protocol based on previously published procedure[56]. Briefly, $5 \times 10^6$ to $1 \times 10^7$ cells were collected and washed with 1 mL cold DPBS buffer, then centrifuged at RT with 500 $g$ to collect the cell pellet. 200 μL of lysis buffer (10 mM Tris-HCl, pH = 7.5, 0.1% NP40, 150 mM NaCl) were added to the cell pellet and incubated on ice for 5 min. The cell lysate was then gently pipetted over 2.5 volumes of chilled sucrose cushion (24% sucrose in lysis buffer) and centrifuged at 4 °C with $15,000 \times g$ for 10 min. Collected all the supernatant as cytoplasmic fraction. Wash the nuclei pellet with 200 μL of ice cold DPBS buffer. The nuclei pellet was resuspended in 200 μL of prechilled glycerol buffer (20 mM Tris-HCl, pH 7.9, 75 mM NaCl, 0.5 mM EDTA, 0.85 mM DTT, 0.125 mM PMSF, 50% glycerol) by gently flicking the tube. An equal volume of cold nuclei lysis buffer (10 mM HEPES, pH 7.6, 1 mM DTT, 7.5 mM $MgCl_2$, 0.3 M NaCl, 1 M urea, 1% NP-40) was then added, followed by vigorous vortexing for 2 × 5 s. Nuclei pellet mixtures were incubated for 2 min on ice, then centrifuged at 4 °C with 15,000 $g$ for 2 min. The supernatant was collected as soluble nuclear fraction/nucleoplasm. The pellet was gently rinsed with cold DPBS without dislodging then collected as chromatin fraction.

## LC-MS/MS

mRNA was purified through two-rounds of poly(A)+ purification and one round of ribosome depletion process by RiboMinus™ Eukaryote System v2 (Invitrogen). Nuclear RNA was treated with two rounds of ribosome depletion process and subjected to size selection by gel shift. Around 50–200 ng RNA was digested by 1 μL nuclease P1 (100U/μL, NEB) in 20 μL 1x reaction buffer at 37 °C for 1 h, followed by the addition of 2.3 μL 10X FastAP buffer (Thermo) and 1 μL FastAP

(1 U/μL, Thermo), then incubated at 37 °C for 1 h. Finally, dilute the sample with 27 μL water. The diluted sample was then filtered (0.22 μm pore size, 4 mm diameter, Millipore), and 10 μL of the solution was injected into the LC-MS/MS. The nucleosides were separated by reverse phase ultra-performance liquid chromatography on a C18 column with online mass spectrometry detection using Agilent 6410 QQQ triple-quadrupole LC mass spectrometer in positive electrospray ionization mode. The nucleosides were quantified by using the nucleoside to base ion mass transitions of 245 to 113.1 (U), 245.1 to 125 (Ψ). Quantification was performed by comparison with a standard curve obtained from pure nucleoside standards run with the same batch of samples. The ratio of Ψ to A + G was calculated based on the calibrated concentrations.

## BID-seq for nuclear RNA

Nucleus fraction was isolated based on cell fractionation protocol. Nuclear RNA was purified by TRIzol reagent (Invitrogen) and ethanol precipitation. rRNA was further removed from Nuclear RNA by RiboMinus™ Eukaryote System v2 (Invitrogen). 50 ng nuclear RNA from *PUS7* KD and control HCT116 cells were used for BID-seq. The BID-seq was performed as previously described[13]. 3 biological replicates were used for both the *PUS7* KD and control conditions.

## PAR-CLIP

We performed PAR-CLIP based on a previously reported protocol[57]. Briefly, four 15 cm plates of HCT116 cells were seeded per replicate and grown to 80% confluency before adding 4 μL of 1 M 4SU to each plate. After 14 h of incubation, the media was aspirated, and the cells were washed once with 5 mL of ice-cold DPBS per plate. The cells were crosslinked on ice using 0.15 J cm$^{-2}$ of 365 nm UV light, applied twice. Crosslinked cells were collected with cell lifters, and four volumes of lysis buffer (50 mM HEPES-KOH pH 7.5; 150 mM KCl, 2 mM EDTA, 0.5% NP-40, freshly added 1:100 protease inhibitor (Roche)) were added to the pellet. After 10 min of incubation on ice with periodic mixing, the lysate was centrifuged at 15,000 x $g$ for 15 min, and the supernatant was collected. RNase T1 (Thermo Scientific) was added to a final concentration of 0.1 U μL$^{-1}$, followed by a 15-min incubation at room temperature, which was then quenched on ice. 5 μg of anti-PUS7 were added to the lysate and incubated at 4 °C for 2 h with periodic rotation. 100 μL Protein A or G beads (Thermo Scientific) were washed twice with IP wash buffer (50 mM HEPES-KOH pH 7.5, 300 mM KCl, 0.05% NP-40, freshly added 1:100 protease inhibitor (Roche) and 40 U/mL SUPERase-In (Thermo Scientific)) and resuspended in 50 μL lysis buffer before being added to the antibody-lysate mixture. Another 2-h incubation at 4 °C with low-speed rotation followed. After incubation, the beads were washed three times with IP wash buffer, resuspended in 200 μL of IP wash buffer, and subjected to a second RNase T1 digestion (10 U μL$^{-1}$ final concentration) for 15 min at room temperature. The reaction was quenched with 10 μL SUPERase-In (Thermo Scientific), followed by a 5-min incubation on ice. Beads were washed three times with high-salt wash buffer (50 mM HEPES pH 7.5, 500 mM KCl, 0.05% NP-40, freshly added 1:100 protease inhibitor (Roche)) and twice with 1x PNK buffer (NEB). They were resuspended in 200 μL of 1x PNK buffer and underwent T4 PNK end repair (Thermo Fisher Scientific) at 37 °C using standard procedures. After incubation, beads were washed once with 1x PNK buffer, followed by proteinase K digestion. RNA was recovered using RNA Clean & Concentrator (Zymo Research) and prepared for library construction using the NEBNext Small RNA Library Prep Set for Illumina (NEB). Alternative adaptors were used for small RNA library preparation. 3′ SR Adapter: 5′App-NNNNNATCACGA-GATCGGAAGAGCACACGTCT-3SpC3; 5′ SR Adapter: 5′-GUUCAGA-GUUCUACAGUCCGACGAUC NNNNN-3′. The libraries were purified on a 3.5% low melting point agarose gel and sequenced on Illumina Novaseq 6000 platform.

## PUS7 expression

Human *PUS7* was cloned into pET28a(+) vector using HiFi DNA Assembly Cloning Kit (NEB). The PUS7 protein was expressed in *E. coli* in LB media. Protein expression was induced at OD600 of 1.2–1.4 with 1 mM isopropyl -D-1-thiogalactopyranoside (IPTG), and cells were harvested after 16–18 h incubation at 20 °C. Cells were lysed by sonication in lysis buffer (50 mM Tris-HCl pH 7.5, 500 mM NaCl). The soluble recombinant protein was purified using a nickel resin column washed with washing buffer (20 mM Tris-HCl pH 7.5, 400 mM NaCl, 30 mM imidazole). The protein was eluted in elution buffer (20 mM Tris-HCl pH 7.5, 400 mM NaCl, 400 mM imidazole) and subjected to HiTrap Heparin HP affinity column on an AKTA Purifier 10 system (GE Healthcare) to get rid of the RNA and DNA bound with the protein. Protein concentrations were determined using Nanodrop. The protein was concentrated using 50 kD MWCO spin filter (Amicon).

## Cell viability

Cell viability was measured using a modified Sulforhodamine B (SRB) colorimetric assay[58]. Briefly, plates were washed with DPBS and fixed with 1× TCA solution (3% trichloroacetic acid in DPBS) for 20 min at room temperature. The wells were then washed five times with tap water and air-dried for 30 min. Next, 1× SRB staining buffer (0.57% SRB in 1% acetic acid aqueous solution) was added, and the plate was incubated for 20 min. Excess stain was removed by washing the wells three times with 1% acetic acid in water. Finally, an elution buffer (0.01 M NaOH in water) was added to solubilize the dye, and the optical density was measured at 490 nm using a microplate reader.

## In-vitro transcription of 7SK

7SK sequence was cloned from total RNA from HCT116 using followed primers: 7SK-RT: AAAAGAAAGGCAGACTGCCACATG; T7-PCR-primer: TAATACGACTCACTATAGGATGT GAGGGCGATCTGGC. The sequence was validated by Sanger sequencing. The 7SK RNA probe was synthesized by T7 RNA Polymerase (NEB) following the manufacture protocol.

## PUS7 catalytic assay

A series of PUS7 amounts (0,2,4,8,16,32,64 pmol) were incubated with 500 ng of in-vitro-transcribed 7SK at 37 °C for 30 minutes in 50 mM Tris-HCl pH 8.0, 70 mM NH₄OAc, 30 mM KCl and 2 mM DTT. Then, the 7SK RNA was purified by RNA Clean & Concentrator Kits (Zymo). The purified RNA was subjected to LC-MS/MS for the measurement of Ψ level.

## PUS7-7SK binding assay

PUS7 ranging from 0 to 500 nM was incubated with 40 nM 7SK probe in binding buffer (10 mM Tris-HCl pH 7.5, 100 mM KCl, 1 mM MgCl₂, 0.1 mM DTT, 5% glycerol) at 37 °C for 30 min. 10 µL of RNA–protein mixture was loaded to the Novex 4–20% TBE gel (Invitrogen) and run at 4 °C for 90 min at 90 V. The gel was stained by 1X SYBR™ Gold Nucleic Acid Gel Stain (Invitrogen). Imaging was carried out by Bio-Rad Molecular Imager FX.

## BIHIND-qPCR assay

The Ψ level of 7SK was measured using the BIHIND-qPCR assay[37]. Briefly, 10 µg of total RNA was required for the assay, with 5 µg used as input and the remaining 5 µg subjected to bisulfite (BS) treatment. For the qPCR reaction, 60 nM of "up probe" for U, 60 nM of "up probe" for Ψ, 60 nM of "down probe" for U, and 60 nM of "down probe" for Ψ were used. The RNA and probes were combined in a total volume of 10 µL and annealed in 1X annealing buffer (10 mM Tris-HCl, pH 7.5, 50 mM NaCl, 1 mM EDTA) through a temperature gradient: 90 °C for 1 min, 80 °C for 1 min, 70 °C for 1 min, 60 °C for 1 min, 50 °C for 1 min, 40 °C for 5 min, and held at 35 °C. Afterward, 10 µL of a mixture containing 0.8 U Bst 2.0 DNA polymerase (NEB, M0537S), 0.1 nmol dATP,

and 2X CutSmart buffer was added, and the reaction was incubated at 45 °C for 20 min. Another 10 µL of a mixture containing 15% PEG8000, 6 mM ATP, 2.5 U SplintR Ligase (NEB, M0375S), and 1X CutSmart buffer was added, and the reaction was incubated at 25 °C for 20 min. The reaction was then diluted with 70 µL of water. qPCR was performed using the Roche LightCycler 96 system.

The estimated Ψ fraction is calculated with the following equation:

$$Estimated\ \Psi\ Fraction = 1 - 0.5^{\Delta\Delta C_t} =$$
$$1 - 0.5 \wedge [(C_t^{BS-treated\ \Psi} - C_t^{BS-treated\ U}) - (C_t^{Untreated\ \Psi} - C_t^{Untreated\ U})]$$

## Immunoprecipitation

HCT116 cells from a 10 cm plate were collected and pelleted by centrifugation at 500 x *g* for 5 min. The cell pellet was resuspended in 5 volumes of lysis buffer (150 mM NaCl, 50 mM Tris-HCl pH 7.5, 1% NP-40, protease inhibitor, 20 U/mL SUPERase-In (Thermo Scientific)) and incubated on ice for 30 min. To remove cell debris, the lysate was centrifuged at 15,000 x *g* for 15 min at 4 °C. 50 µL aliquot of the cell lysate was reserved as the input sample, while the remaining lysate was incubated with anti-LARP, anti-HEXIM1 or anti-MEPCE for 2 h at 4 °C. Protein A beads (Invitrogen) were washed with lysis buffer and added to the mixture for an additional 2-h incubation at 4 °C. The beads were then washed three times with ice-cold lysis buffer. The immunoprecipitated samples were used for western blotting or RNA extraction alongside the input samples.

## Immunofluorescence Staining

HCT116 cells were seeded at 50% confluency on 8-well chamber slides 12 h prior to *PUS7* knockdown. After 3 days of *PUS7* knockdown, cells were washed once with DPBS and fixed with 4% paraformaldehyde for 15 min at room temperature. Cells were then washed three times with DPBS and permeabilized with 1% (v/v) Triton X-100 in DPBS for 10 min. After three additional washes with DPBS, cells were blocked with 1% BSA in PBST for 1 h at room temperature. Primary antibodies (rabbit anti-S2P and mouse anti-Pol II, 1 µg/mL each) in PBST were applied, and slides were incubated overnight at 4 °C. The next day, cells were washed three times with PBST, followed by incubation with secondary antibodies (1 µg/mL) for 1 h at room temperature. After three additional PBST washes, cells were stained with 1 µg/mL DAPI for 5 min. Images were acquired using a Leica Stellaris 8 Laser Scanning Confocal Microscope, and fluorescence intensity was quantified using ImageJ software.

## m⁶A-RIP-qPCR

We performed m⁶A-MeRIP enrichment followed by RT-qPCR to quantify the relative m⁶A methylation level or level changes of 7SK. 10 µg of purified total RNA extracted from HCT116 cells was fragmented in 1x zinc fragmentation buffer (10 mM ZnCl₂, 10 mM Tris-HCl pH 7.5) and quenched with an equal amount (relative to ZnCl₂) of EDTA, pH 8.0. For m⁶A-MeRIP, 1 µL of anti-m⁶A antibody (NEB) was incubated with 20 µL of washed Dynabeads Protein G beads (Thermo Fisher Scientific) in 200 µL of IP buffer (10 mM Tris-HCl pH 7.5, 150 mM NaCl, 0.1% NP-40) at 4 °C for 1 h. The beads were then washed three times with IP buffer and resuspended in 200 µL of IP buffer. To the fragmented RNA, 1 µL of 1:100 diluted A-probe and m⁶A-probe from the EpiMark *N*⁶-Methyladenosine Enrichment Kit (NEB) was added, and this mixture was subsequently combined with the antibody-bound beads. The RNA-bead suspension was incubated with rotation at 4 °C for 2 h. Following incubation, the beads were washed twice with IP buffer and twice with HS buffer (10 mM Tris-HCl pH 7.5, 500 mM NaCl, 0.1% NP-40). m⁶A-enriched RNA was finally eluted with 25 µL of Buffer RLT (Qiagen), recovered using an RNA Clean & Concentrator kit (Zymo), reverse transcribed with PrimeScript RT Master Mix (Takara), and then

subjected to RT-qPCR. A spike-in control was used for normalization during qPCR.

## KAS-seq

KAS-seq was performed following a previously reported protocol[44,45]. Briefly, HCT116 cells were transfected with siPUS7, 7SK ASO, or siCtrl and harvested into 1.5 mL tubes after 3 days. Cells were suspended in 1 mL of RPMI medium containing 500 mM N3-kethoxal and incubated on a shaker at 37 °C for 10 min to label genomic ssDNA. The medium was then discarded, and genomic DNA was extracted. The N3-kethoxal-labeled ssDNA was biotinylated, enriched, and fragmented according to the established KAS-seq protocol. Dual-index libraries were prepared for high-throughput sequencing using the Accel-NGS Methyl-seq DNA Library Kit and sequenced on Illumina Nova-seq 6000 platform.

## mRNA-seq

HCT116 cells were transfected with siPUS7, 7SK ASO, or siCtrl and harvested into 1.5 mL tubes after 3 days. mRNA was extracted and applied to mRNA-seq using NEBNext® Ultra™ II RNA Library Prep Kit (NEB). The libraries were sequenced on Illumina Nova-seq 6000 platform.

## ChIP-qPCR

Cells were collected and crosslinked with 1% formaldehyde in the culture medium for 10 min, and the reaction was quenched with 125 mM glycine. The cells were then lysed in ChIP buffer (50 mM HEPES KOH pH 7.5, 1% Triton X-100, 0.1 % sodium deoxycholate, 0.1% SDS, protease inhibitor) and the chromatin was sheared by sonication to a size range of 200–1000 bp. 50 μL of sheard chromatin was saved as input. The remaining sheared chromatin is incubated with anti-Pol II antibody at 4 °C overnight. Protein A beads (Invitrogen) were washed with ChIP buffer and added to the mixture for an additional 2-hour incubation at 4 °C. The beads were then washed three times with ice-cold ChIP buffer and once with LiCl wash buffer (0.25 M LiCl, 1% NP-40, 1% sodium deoxycholate, 10 mM Tris-HCl pH 7.5). The washed beads were mixed with elution buffer (100 mM NaHCO$_3$, 1% SDS, 250 mM NaCl, 200 μg/mL RNase A (Invitrogen), 1 mg/mL protease K) and incubated at 60 °C for more than 4 h. The DNA was purified by ethanol precipitation along with input. qPCR is performed with primers targeting promoter and gene body regions to quantify the ratio of Pol II occupancy in the gene body over the promoter.

## Flow cytometry

HCT116 cells were collected and washed with DPBS, then resuspended in Annexin V Binding Buffer (Biolegend) at a density of $1–5 \times 10^6$ cells/mL. To the cell suspension, 5 μL of FITC-conjugated Annexin V (Biolegend) was added to 100 μL of the cell suspension, followed by incubation for 15 min at room temperature, protected from light. After incubation, 2 mL Annexin V Binding Buffer was added, and the cells were centrifuged at $600 \times g$ for 2 min at room temperature. The supernatant was discarded, and the cells were resuspended in 200 μL Annexin V Binding Buffer. To assess cell viability, 5 μL of Propidium Iodide Solution (Biolegend) was added, and the cells were incubated for 15 min at room temperature. The samples were analyzed by flow cytometry.

## Bioinformatics

The codes for sequencing data analysis have been publicly available on GitHub (https://github.com/YutaoZhao1224/PUS7-project).

BID-seq: Only R1 reads of paired-end reads were processed for BID-seq data analysis. R1 was trimmed by cutadapt[59] (version 4.8) with following parameters "-a ATCACGAGATCGGAAGAGCA -g TTCTA-CAGTCCGACGATC -e 0.1 -q 20 -m 32 -n 3 -O 8" to remove adaptors and

low-quality reads. PCR duplicates were removed with the BBMap tool (v.38.73.). 5-mer random barcodes at reads ends were trimmed by cutadapt[59] (version 4.8). Remaining reads were aligned to hg38 by STAR mapping software[60] (2.7.11b). The generated bam file was sorted by samtools[61] (1.21) and subjected to samtools mpileup for searching deletion signature. A Ψ candidate was required to satisfy the following criteria: (1) deletion ratio above 0.05 (with deletion count above four) in bisulfite-treated library; (2) deletion ratio below 0.02 in input libraries; (3) total read coverage depth above 20 for bisulfite-treated and input libraries; (4) all deletion signatures must be from 'U' sites marked by hg38; (5) all deletion signatures should be identified in each biological replicates. The differentially regulated Ψ sites were identified by two-tailed Student's $t$ test ($p < 0.05$).

PAR-CLIP: Only R1 reads of paired-end reads were processed for PAR-CLIP data analysis. R1 was trimmed by cutadapt[59] (version 4.8) with following parameters "-a ATCACGAGATCGGAAGAGCA -g TTCTACAGTCCGACGATC -e 0.1 -q 20 -m 32 -n 3 -O 8" to remove adaptors and low-quality reads. PCR duplicates were removed with BBMap tool (v.38.73.). 5-mer random barcodes at reads ends were trimmed by cutadapt[59] (version 4.8). Remaining reads were aligned to hg38 by bowtie2 mapping software (2.4.4). The aligned sam files were subjected to PARalyzer[62] (v1.5) for peak identification. The parameters in the ".ini" file used as the input for PARazyler are as follows:

```
BANDWIDTH=3
CONVERSION=T>C
MINIMUM_READ_COUNT_PER_GROUP=5
MINIMUM_READ_COUNT_PER_CLUSTER=2
MINIMUM_READ_COUNT_FOR_KDE=3
MINIMUM_CLUSTER_SIZE=11
MINIMUM_CONVERSION_LOCATIONS_FOR_CLUSTER=2
MINIMUM_CONVERSION_COUNT_FOR_CLUSTER=3
MINIMUM_READ_COUNT_FOR_CLUSTER_INCLUSION=1
MINIMUM_READ_LENGTH=20
MAXIMUM_NUMBER_OF_NON_CONVERSION_MISMATCHES=1
```

KAS-seq: The paired-end reads were trimmed by cutadapt[59] (version 4.8) with following parameters "-a AGATCGGAAGAGCA-CACGTCTG -A AGATCGGAAGAGCGTCGTGT -n 3 --pair-filter=any -n 3 -O 8" to remove adaptors and low-quality reads. PCR duplicates were removed with BBMap tool (v.38.73). Remaining reads were aligned to hg38 by bowtie2 mapping software (2.4.4). The generated bam file was sorted by samtools[61] (version 1.21). The sorted bam files were converted to bigwig files by bamCoverage[63] (deeptools 3.5.5). The peaks were identified by MACS2[64] (version 2.2.6) and visualized by IGV[65] (version 11.0.13). The read coverage on promoter (from TSS-100 to TSS + 300 bp) and gene body (from TSS + 300 to TSS + 2000 bp) was calculated by bedtools coverage[66] (v2.30.0). The Pol II release ratio was calculated as read coverage of gene body over promoter. Those genes with promoter read coverage less than 20 were discarded for data analysis.

RNA-seq: The paired-end reads were trimmed by cutadapt[59] (version 4.8) with following parameters "-a AGATCGGAAGAGCA-CACGTCTG -A AGATCGGAAGAGCGTCGTGT -e 0.15 -q 20 --nextseq-trim=20 -O 6 -n 3 --pair-filter=any" to remove adaptors and low-quality reads. PCR duplicates were removed with BBMap tool (v.38.73). Remaining reads were aligned to hg38 by STAR mapping software[60] (2.7.11b). The generated bam file was sorted by samtools[61] (version 1.21). FeatureCounts[67] (version 2.0.3) was used to count reads mapped to protein-coding genes from hg38, and differentially expressed gene analysis was conducted by DEseq2[68] software. Differentially expressed genes were identified with a cutoff of p-adj < 0.01, $|\log_2 FC| \geq 1$.

## Reporting summary

Further information on research design is available in the Nature Portfolio Reporting Summary linked to this article.

## Data availability

The data supporting the findings of this study are available from the corresponding author upon request. The BID-seq, PAR-CLIP, KAS-seq and mRNA-seq data have been deposited at the Gene Expression Omnibus (GEO) under the accession number GEO: GSE288120. The processed data for BID-seq, PAR-CLIP, KAS-seq and mRNA-seq are provided in Supplementary Data 1–4. Details of the primers and siRNAs used in this study are also included in Supplementary Data 5. Source data are provided with this paper.

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

## Acknowledgements

We thank members of the He lab for critical discussions. We thank Dr. Pieter W. Faber and Genomics Facility of the University of Chicago for their generous help with high-throughput sequencing. This work was supported by the US National Institutes of Health (R01HL155909 and RM1HG008935 to C.H.) and the Josef Fried Chemical Biology Fellowship (Y.Z.). C.H. is an investigator of the Howard Hughes Medical Institute.

## Author contributions

Y.Z. and C.H. conceived the project. Y.Z. planned, designed and executed experiments, analyzed the data and produced figures with the help from H.-L.S., W.L., T.W., X.Y., P.W., C.-W.J., Y.Z., Q.D., and K.P. Y.Z. analyzed high throughput data and performed bioinformatic analyses with the help from C.Y., S.L., and Y.P. Y.Z. and C.H. wrote the manuscript with helpful discussion from H.-L.S. and X.D.

## Competing interests

C.H. is a scientific founder, a member of the scientific advisory board and equity holder of Aferna Bio, Inc. and Ellis Bio, Inc., a scientific cofounder and equity holder of Accent Therapeutics, Inc., and a member of the scientific advisory board of Rona Therapeutics and Element Biosciences. The remaining authors declare no competing interests.
