## [Transparent Peer Review file · Nature Communications]

Pseudouridylation of 7SK by PUS7 regulates Pol II transcription elongation

Corresponding Author: Professor Chuan He

Version 0:

Reviewer comments:

Reviewer #1

(Remarks to the Author)

Although RNA pseudouridylation plays a crucial role in RNA metabolism by regulating RNA splicing and mRNA translation, its function in controlling transcription remains unclear. In this manuscript, Zhao et al. demonstrate that pseudouridylation inhibits global cellular transcription by promoting RNA Pol II promoter-proximal pausing. Mechanistically, pseudouridine synthase 7 (PUS7) mediates the pseudouridylation of 7SK snRNA, which in turn prevents the release of the p-TEFb complex from 7SK snRNA. As a result, transcription elongation is inhibited, leading to the suppression of global cellular transcription.

More importantly, the authors observed that the removal of pseudouridine modification from 7SK facilitates KLF6/DDIT3-mediated cell apoptosis and enhances the sensitivity of colorectal cancer (CRC) cells to 5-FU treatment. Overall, this study provides compelling evidence that not only elucidates the role of pseudouridylation in global transcriptional regulation but also suggests that targeting PUS7 may sensitize CRC cells to 5-FU, thereby offering a novel therapeutic strategy for CRC treatment. Nonetheless, addressing the following specific points can improve this manuscript.

Major Concern

1. Throughout the manuscript, it is unclear if PUS7 is responsible for only a single ψ site (ψ 250) or there are multiple ψ sites on 7SK. For example, in Fig. 1K, four ψ sites in what appear to be different 7SK genes decrease after PUS7 knockdown. Do all four ψ sites correspond to nt 250? Besides, 3 of the 4 7SK sites noted are in 7SK pseudogenes and not in the real genomic copy of 7SK? Fig. 2C implies that there are at least two ψ sites on 7SK. In addition, it would be helpful to determine ψ sites in the in vitro pseudouridylated 7SK and check if it is consistent with 7SK in cells. Finally, if there are other ψ sites, it would be useful to determine if they are affected by the dCas13b-PUS7 in Fig. 5, and whether they also contribute to 5-FU sensitivity.
2. The authors show that pseudouridylation of 7SK increases HEXIM1 binding to 7SK but fail to observe the same increased pseudouridylation of 7SK ψ 250 upon CDK9 RIP. They should provide a justification for this result that explains the role of ψ 250.
3. The gating strategy in both siCtrl and siPUS7 in Supplemental Figure 1D appears to be inappropriate. The authors should reanalyze this data to ensure accuracy.
4. While the authors evaluate the effect of PUS7 knockdown on LARP7, HEXIM1, and MEPCE interactions with P-TEFb components, they should also demonstrate the effect that this knockdown has on hnRNPs that are known to bind to 7SK during active transcription.
5. In Figures 3F and 3G, the authors show that CDK9, HEXIM1, and 7SK all shift towards the smaller fractions of a glycerol gradient. However, this does not provide justification that these components dissociate from each other, as they still sediment in the same fraction.

Minor concerns:

1. In Figure 1F, the authors claim there is no considerable change in pseudouridylation of nuclear RNA <200 nt in length. Given the difference is statistically significant, the authors should provide a stronger justification for not acknowledging this difference.
2. In Figure 2D, the authors show the PUS7 PAR-CLIP peaks, but these peaks are mainly at the very ends of the molecule and not near the ψ 250 site. The authors should clarify whether there is a known PUS7 binding motif and provide justification for why this site classifies as overlapping.
3. In Figures 2H-J, the authors calculate a pseudouridine level at 7SK site ψ 250 using the BIHIND-qPCR method. However, they do not describe in the corresponding methods section how they calculated this value.
4. In Figure 4A and the corresponding methods section, the authors need to clarify what normalization is done to their KAS-seq data. As it currently reads, the coverage is a maximum of 8 reads across 7962 genes.
5. In the text, the order of some figures is incorrect, such as: in line 149, 'Fig. S4A' should be Fig. S3E. in line 226, 'Figure 5B' should be corrected to 'Figure 4M'. In line 228 "Figure 5C" should be updated to "Figure 4N". in line 233 "Figure S4D" should be 4I and 4J.
6. According to the data presented in Figure 5C, the knockdown of either KLF6 or DDIT3 completely rescues the cell growth inhibition caused by PUS7 knockdown. However, in the cell apoptosis analysis shown in Figures 5D and 5E, PUS7 knockdown-induced apoptosis is only partially rescued by knocking down either KLF6 or DDIT3. Could the authors provide an explanation for this discrepancy?
7. The data presented in Figure 5F indicate a negative correlation between PUS7 and KLF6 expression. The authors should also provide the expression correlation between PUS7 and DDIT3.
8. Previous studies have suggested that high DDIT3 expression sensitizes CRC cells to 5-FU treatment. Given that PUS7 depletion upregulates DDIT3 expression, the authors confirmed that PUS7 knockdown significantly enhances the 5-FU sensitivity of CRC cells. However, since PUS7 is highly expressed in CRC patients, its overexpression, but not elevated ψ levels, in CRC cells may contribute to 5-FU chemoresistance. Therefore, the authors should consider testing or predicting whether pharmacological inhibition of PUS7 with a small molecule could sensitize CRC cells to 5-FU treatment.
9. According to the provided data, PUS7 knockdown results in a 30% to 40% reduction in pseudouridylation at U250 of 7SK. A published study suggests that DKC1 also mediates pseudouridylation at this site. To further investigate this mechanism, the authors should examine whether the simultaneous depletion of PUS7 and DKC1 can completely eliminate pseudouridylation on 7SK snRNA in CRC cells.
10. There are many typos and grammatical errors that should be corrected prior to publication, including in the first sentence of the paper: line 11 "pseudouridine is 'a' widespread RNA modification...".
11. Legend of Fig. S1 should include HT-29 cells.

Reviewer #2

(Remarks to the Author)

Reviewer #3

(Remarks to the Author)

Reviewer #4

(Remarks to the Author)

Reviewer #5

(Remarks to the Author)

This study by Zhao et al examines the role of 7SK pseudouridylation on RNA Pol II promotor pausing and elongation. Importantly, it identifies PUS7 as the pseudouridine synthetase responsible for pseudouridylating 7SK at position 250 and

brings to light how this impacts downstream processes. The paper is well structured and well written and provides important new knowledge. I nonetheless have several comments that I would ask the authors to address.

1. I find the title slightly misleading. While it is clear that pseudouridylation of 7SK is important in regulating the release from proximal promoter pausing, it is not the only effector as the authors also acknowledge the impact of m6A. Given that 7SK pseudouridylation appear co-transcriptional and m6A installation is post-transcriptional (<https://www.biorxiv.org/content/10.1101/2025.03.07.641986v1>), it is not clear whether the absence of pseudouridine itself is enough to drive the observed effect or whether m6A installation is still crucial. I would suggest that repeating select experiments using PUS7 silencing in the presence of a METTL3 inhibitor (e.g. STM2457) would shed further light on the exact importance of the two modifications.

2. The data presented convince me that PUS7 is capable of pseudouridylating 7SK and mostly likely does so in cellular contexts. However, given that this role was previously considered to be performed by DKC1 (line 50), I am surprised the authors did not try to silence DKC1 expression to verify that 7SK pseudouridylation at position 250 was not affected.

3. The last paragraph of the discussion is jarring and somewhat out of context given the focus of the manuscript on pseudouridine. In general the discussion could be rewritten for better clarity and the inclusion of m6A would be enhanced by the inclusion of the experiment suggested in point 1 above.

Other comments

1. As written, I find the title slightly misleading (see point 1 above)

2. Given that the introduction suggests DKC1 is likely responsible for pseudouridylating 7SK (line 50), I do not understand why the authors considered PUS7 to be an alternative candidate. While they are obviously proven right by their work, it would be worth explaining why they started with PUS7 (specifically).

3. Lines 89-91 and Figure 1. Both panel 1G and 1F show significant decreases and it is not clear to me what one is considerably more than the other. The authors should be clearer about this.

4. Line 113. I'm not super experienced with PAR-CLIP but I don't find these results to be highly consistent. Certainly, there is a reasonable overlap but not much more than that. I also wonder why none of the other PUS7-dependent pseudouridine sites were identified here?

5. Line 119. Several studies have now shown additional pseudouridine sites in 7SK (e.g. 242, 247). Where these also observed here?

6. Line 178. It's quite a leap to state that the lack of change in the stability of the BAF complex equates with a lack of change in function. This should either be tested experimentally or else carefully reworded.

7. Line 226 and 228 – References should be to supplementary and not main figures.

Reviewer #6

(Remarks to the Author)
(attached)

In this manuscript, Zhao and colleagues report on how the pseudouridine modification may regulate transcription via PUS7-mediated pseudouridylation of 7sK snRNA, which regulates Pol II pausing. They demonstrate that a reduction of PUS7 leads to a reduction of psi occupancy on 7SK, which leads to dissociation of P-TEFb from 7SK and enhances transcription elongation. This topic is of great interest for the RNA modification community, as many of the current studies are focused on the effects of pseudouridylation on translation. The possible connection with transcription is exciting. I have several comments on how to improve the manuscript.

Major comments:

1. The protein knockdown in panel S1A looks very minimal, which is concerning for interpreting the results. Please provide qRT-PCR to demonstrate the level of RNA knockdown that was achieved after 3 days (according to the method) and also quantify the level of KD by western blot. Additionally, the partial knockdown should be elaborated on in the discussion section and how this might influence the observations.

2. Considering the partial knockdown of Pus7, it is hard to believe the quantification in Figure 1E—is it possible that the changes were minimal due to incomplete knockdown and detection limits of the instrument? The detection limit should be tested, and these limitations should be mentioned in the discussion.

3. Line 99: 42 identified sites should be orthogonally validated by another method and verified to not be genomic SNVs at those positions. Please elaborate on the filtering criteria for determining differences in psi levels (read cutoff, minimum occupancy levels, etc.) in the text. For the sites that had differences, are there significant differences in the RNA expression levels?

4. Line 113-114: The replicate PAR-CLIP experiments do not look highly consistent with each other; the overlap seems 50/50. More replicates needed.

5. Line 140: How did the RNA expression levels change for 7SK for each of the mentioned cell types? This should be used in the interpretation of the psi fraction (deletion ratio)

6. Line 174: The Ser2 phosphorylation difference is visually subtle; please validate using an orthogonal method, as this is supporting a central claim. Also, please quantify all bands, not just the Ser2 phosphorylation.
7. Line 210: It is unclear how to interpret Figure 4H because the ASO has much higher expression for ERG1 and DDIT3 and very modest enhancement for siPUS7
8. Line 220: principal component analysis, which accounts for 50% of the variance—this makes it a bit of an overstatement to say that PUS7- and 7SK-deficient cells share similar transcriptomes. Also, the gene expression differences were quite significant in the three examples in Figure 4H. The language about these samples being similar should be softened, or a more convincing demonstration is needed that these cells are similar.
9. Line 272: In Figure 5J I can see that the relative psi level increased, but what is the starting level of the negative control? Is it close to zero? Is it possible that the gRNA binding to 7SK leads to the observed phenotype? This could be tested by using the same gRNA not fused to PUS7 and adding it to the Figure 5J-M plots.

Minor comments:

1. In the introduction, the statement that psi is the most abundant RNA modification is debatable. It would be helpful for readers to provide the actual numbers and relative to m6A with citations instead.
2. On line 45, it would be helpful to the reader to have a short definition of proximal pausing.
3. Line 73-74: quantify “significantly upregulated”. Also, please elaborate in the text whether GTEX can distinguish Pus7 from Pus7L.
4. Line 81: It would be more accurate to say “...located in the nucleus but not bound to chromatin”
5. Line 128-130: It would be easier to evaluate the Kd if a control RNA (perhaps with the same motif?) that is not a natural substrate for PUS7
6. Line 135: I would appreciate if psi level were referred to as the deletion ratio throughout the manuscript
7. Line 155: Quantify “substantially reduced”. Also, please include absolute values in the text or the supplement, as small numbers can have large relative differences, but the interpretation is not the same. (Fig 3C and D)
8. Figure 3F: The shift seems rather subtle, please include the number of replicates in the figure legend to make this more convincing. Also, the bands seem like the alignment in the image is a little off, please check.
9. Line 213: Please include here the number of genes contributing to the specific apoptosis-related enrichments.

Version 1:

Reviewer comments:

Reviewer #1

(Remarks to the Author)

Our major concerns were largely alleviated by the responses of the authors. However, we want to advise adding some data and discussions in the “responses to reviewers” document to the revised manuscript.

1. In response to Reviewer 1, Major point 1: While the authors acknowledge the genomic copy of 7SK is the chromosome 6 positions and the other copies of 7SK sequence are pseudogenes, they should update manuscript Figure 1K to more accurately reflect the 7SK pseudogenes as such. Labeling these columns just as they are labeled in Figure 1 of the reviewer response would suffice. Additionally, the multiple sequence alignment in Figure 1 from the reviewer response should be added to the supplement to bolster the claim that this modification occurs at the conserved 250 position of 7SK. Finally, the notion that U250 is the only pseudouridylated site should be mentioned early in the manuscript, when describing Figure 1K.
2. In response to Reviewer 1, Minor point 2: The authors expanded on their description of PUS7 interactions with RNA substrates. While there is an extensive description that is included in the response to reviewers, including more of this information in the text of the article would bolster their argument that the PUS7 binding can be far away from the pseudouridylation site.
3. In response to Reviewer 1, Minor point 6: The authors note that while KLF6 and DDIT3 knockdowns restore cell proliferation, they also stimulate apoptosis by other mechanisms. Discussion of this proposed mechanism should be included in the article.
4. In response to Reviewer 1, Minor point 7: The authors note and show in the figure response that the correlation between PUS7 and DDIT3 is positive. This result should be included in either the supplemental or main figures and the discussion justifying this difference as compared to KLF6.
5. In response to Reviewer 2, Major point 2: While the authors acknowledge that the DKC1 knockdown had limited impact

on pseudouridylation at site 250, this information and the corresponding figures (Figures 2A and 2B for Reviewer 2) should be included in the text to provide context for why it was not necessary to knock DKC1 down.

Reviewer #2

(Remarks to the Author)

Reviewer #3

(Remarks to the Author)

Reviewer #4

(Remarks to the Author)

Reviewer #5

(Remarks to the Author)

The authors have satisfactorily responded to my critiques. This remains a very interesting and exciting paper!

Reviewer #6

(Remarks to the Author)

The authors have addressed all of my concerns.
